# Late Neogene nannofossil assemblages as tracers of ocean circulation and paleoproductivity over the NW Australian shelf

Boris-Theofanis Karatsolis[1] and Jorijntje Henderiks[1]

[1]Department of Earth Sciences, Uppsala University, Uppsala, 752 36, Sweden

*Correspondence to*: Boris Theofanis Karatsolis (boris.karatsolis@geo.uu.se)

**Abstract.**

Late Miocene to Pliocene sediments from the NW Australian shelf, drilled by the International Ocean Discovery Program (IODP) Expedition 356, provide unique records of paleoclimatic variations under warmer-than-present conditions. During the period from 6-3.5 million years ago (Ma), the area was dominated by warm, tropical waters supplied by an intensified,

uninterrupted Indonesian Throughflow and characterised by prevailing humid conditions, including increased precipitation and river runoff. Despite the available information regarding the general paleoclimatic conditions, little is known about the concurrent regional ocean circulation patterns and the relative strength of seasonally flowing boundary currents, such as the Leeuwin Current. In this study, we investigate two astronomically-tuned calcareous nannofossil time series from IODP Sites U1463 and U1464 to track long-term changes in ocean circulation and water column stratification, which influences the

availability of nutrients in the upper photic zone and is considered a primary control on the (paleo)productivity of marine phytoplankton. By documenting shifts in the dominant species within the nannofossil assemblages and comparing these to paleotemperature gradients between the NW Australian shelf and the eastern Indian Ocean, we identify a significant change in ecological and oceanographic regime that occurred across the Miocene-Pliocene boundary (5.4–5.2 Ma), which can be attributed to an overall intensification of the upper water column mixing over the shelf. Significant changes in nannofossil

abundance and species composition that reflect broader-scale processes and evolutionary events, such as the termination of the late Miocene to early Pliocene biogenic bloom in the eastern Indian Ocean (4.6-4.4 Ma) and the extinction of *Sphenolithus* spp. (~3.54 Ma), occurred long after this regional regime shift.

## 1 Introduction

Calcareous nannoplankton (including coccolithophores) are sensitive indicators of current and past climatic change, because as primary producers, they are primarily dependent on the prevailing (and seasonally changing) physicochemical conditions, such as nutrient availability and temperature, within the upper sunlit layers of the ocean. The abundant and well-preserved fossil record of their calcareous platelets (calcareous nannofossils) in shallow marine and deep-sea sediments has been used to reconstruct regional paleoclimatic and paleoceanographic changes, but also global processes such as evolution (speciation and extinction) and basin-wide changes in ocean chemistry (Beaufort et al., 1997; Bolton et al., 2016; O'Dea et al., 2014; Thierstein and Young, 2004). Because of the interplay of regional and global imprints preserved in the sedimentary record, nannofossil assemblage studies come with a set of challenges, especially in highly dynamic environments across continental shelves. In such areas, calcareous nannoplankton communities may be significantly different from open ocean marine settings, owing to the influence of parameters such as seasonal water column stratification and tidal mixing (Sharples et al., 2009; Van Oostende et al., 2012), as well as variations in river runoff and nutrient input (Harlay et al., 2010; Poulton et al., 2014). This further complicates the task of using nannofossil assemblages to consistently differentiate between signals of regional paleoclimatic forcing and broad-scale evolutionary trends and changes in the ocean's nutrient budget.

The NW Australian shelf serves as an important test-case to try and tackle these challenges, because of its position within the dynamic climate system of the Indo-Pacific warm pool and near the only remaining "equatorial warm water valve". Modern ocean circulation in the area is controlled by a variation in intensity of the Indonesian Throughflow (ITF) and the Leeuwin Current (LC). The ITF is the warm surface current that transports water from the Western Pacific Warm Pool (WPWP) into the Indian Ocean (Du and Qu, 2010; Gordon et al., 1997; Figure 1). The LC is a boundary current that is sourced by the ITF and transports warm water southwards (Figure 1), with significant implications for fisheries and the persistence of tropical biota in the high latitudes of western and southwestern Australia (e.g. Kendrick et al., 2009 and references therein). Its strength varies seasonally, with the strongest flow being observed during austral winter, when latitudinal steric height gradients are steeper (e.g. Godfrey and Ridgway, 1985; Ridgway and Godfrey, 2015). Interannually, stronger/weaker LC is linked to La Niña/El Niño conditions, with a mean annual transported volume of 4.2 Sv/3 Sv (e.g., Feng et al., 2003, 2009). Additionally, this flow is known to consist of warm-core anticyclonic eddies that can promote phytoplankton production across the western Australian shelf (Koslow et al., 2008; Thompson et al., 2007). The NW Australian shelf is a source area of the LC, contributing to its total volume through a surface current called the Holloway Current (HC; D'Adamo et al., 2009; Holloway and Nye, 1985), that flows south-westward along the shelf. Weak upwelling events have been observed in the area both during summer and winter months, when westerly winds are strong enough to overcome the steric height gradient (Holloway and Nye, 1985). Modern observational data and forecasts (E.U. Copernicus Marine Service Information) show that during austral winter, when the LC is strong, the NW Australian shelf area is characterised by a mixed layer deepening, which reaches ~100m. At the same

time, chlorophyll-a increases in the surface layer, driven by an increase in water column cooling and advective mixing (Figures 2, 3).

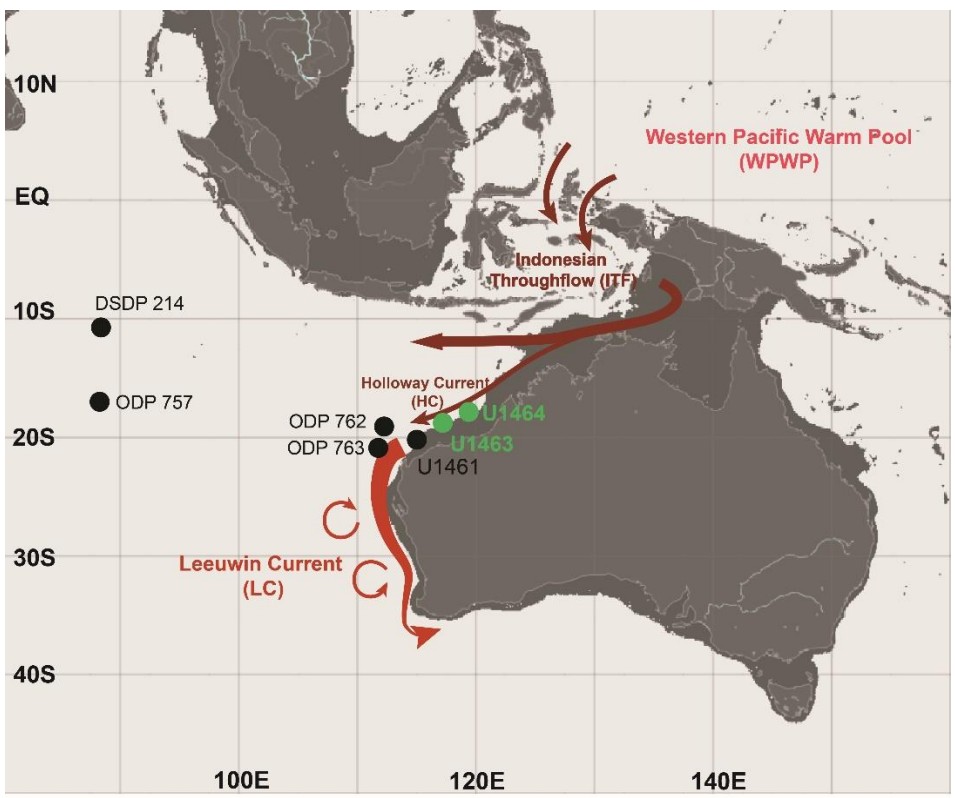

**Figure 1** Modern bathymetric map indicating shelf environments (<200 m depth; dark grey shading) and site locations. IODP Sites U1463 and U1464 (NW Australian shelf; green dots) are the primary sites of investigation in this study. Other sites discussed in the text (black dots) include IODP Site U1461 (NW Australian shelf), ODP Sites 762 and 763 (Eastern Indian Ocean), as well as ODP Site 757 and DSDP Site 214 (equatorial Indian Ocean). The main surface oceanography of the Indo-Pacific region, including the ITF (dark red arrows) and main path of the LC (lighter red arrow; adapted from Auer et al., 2019; Gallagher et al., 2009) and the HC, which in this study are considered as one, are shown. The base map was generated using the ocean data visualization tools of E.U. Copernicus Marine Service Information (MYOCEAN PRO).

Information on the existence and strength of the LC in the latest Miocene and Pliocene remains limited, although a possibly active LC has been inferred from upper water column temperature and primary productivity reconstructions at its source area (IODP Expedition 356 Site U1461; Figure 1) for the interval spanning from 6-4 Ma (He et al., 2021). During that time, increased river runoff and humid conditions were prevailing in the area, under an intensified Indonesian Throughflow (ITF) (Auer et al., 2019; Christensen et al., 2017; He et al., 2021; Karatsolis et al., 2020). Previous investigations have demonstrated that nannofossil assemblages can reflect ITF source water variations and LC activity during the late Pliocene to Pleistocene (Auer et al., 2019), as well as record broader changes in the ocean's nutrient budget. Evidence for the latter comes from the significant decrease in fluxes of the opportunistic, fast-growing (bloom-forming) species within the assemblages, from 4.6–

4.4 Ma, which marked the end of the late Miocene to early Pliocene biogenic bloom over the NW Australian shelf (Karatsolis et al., 2020; 2022).

In this study, we aim to further constrain the latest Miocene to early Pliocene ITF and LC history, by identifying the key changes in the calcareous nannofossil assemblages. To achieve that, we extend the nannofossil database from Site U1463 (Auer et al., 2019; 3.65-2.97 Ma) back to ~5.7 Ma and present one more astronomically-tuned record from Site U1464, that is located ~100 km away but in a different basin, dating back to ~6 Ma (Gallagher et al., 2017a; Karatsolis et al, 2020). We evaluate the use of a ratio between three abundant taxonomic groups (genera) as an index to trace long-term shifts in water-column stratification and regional ocean circulation. Comparison with newly constrained temperature gradients between the NW Australian shelf and the eastern Indian Ocean further assists in identifying the possible underlying mechanisms for the observed changes. Finally, nannofossil evidence is put in a broader perspective in an effort to differentiate between signals that reflect basin-wide or global changes in nannoplankton evolution and the ocean's nutrient budget, and those that occurred due to regional ocean circulation and intensity of seasonal variations. Such decoupling of regional signals from overlying global patterns is crucial when using nannofossil assemblages to reconstruct the evolution of the LC, as well as other boundary currents and ocean circulation patterns in dynamic shelf environments.

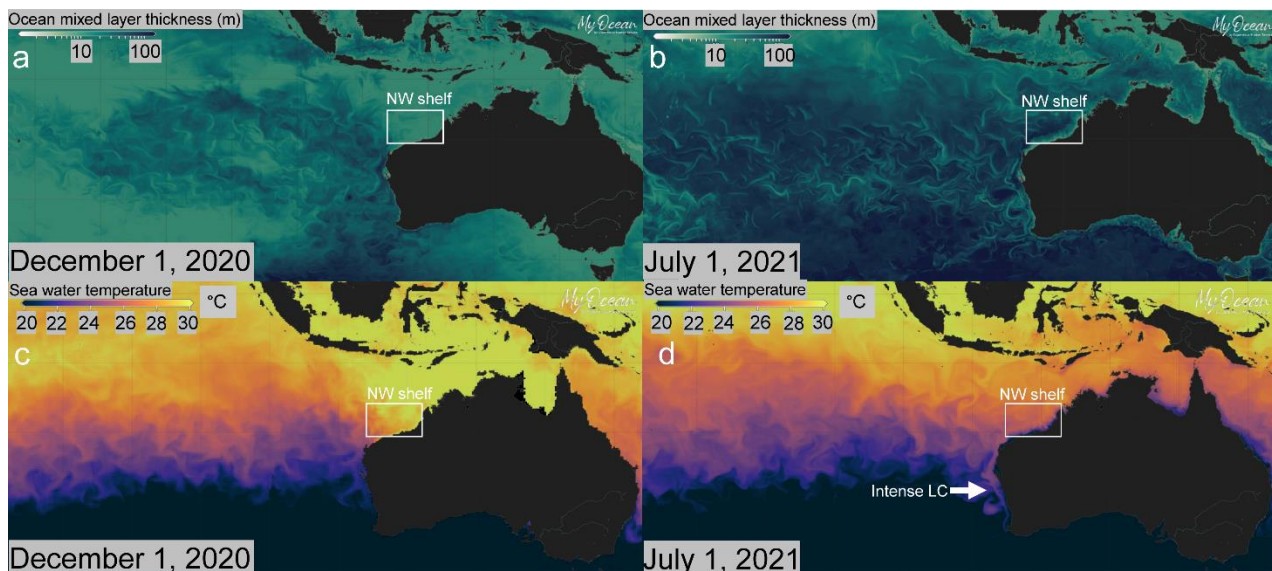

**Figure 2** Present-day ocean circulation maps of the southeastern Indian Ocean area for the 1st of December 2020 (austral summer; **a, c**) and 1st of July 2021 (austral winter; **b, d**). LC is weak during austral summer causing **(a)** reduced depth of the mixed layer and increased stratification and **(c)** higher SST over the NW Australian shelf. The LC intensifies during austral winter, when **(b)** water column mixing in the eastern Indian Ocean increases and **(d)** SST decreases in the NW Australian shelf area; at the same time, it transports warmer surface waters further to the south (d). The sea surface cooling effect observed in shelf environments during July is less pronounced further offshore. Maps were generated using the ocean data visualization tools of E.U. Copernicus Marine Service Information (MYOCEAN PRO) and dataset layer GLOBAL_ANALYSIS_FORECAST_PHY_001_024 (https://doi.org/10.48670/moi-00016) and GLOBAL_ANALYSIS_FORECAST_BIO_001_028 (https://doi.org/10.48670/moi-00015).

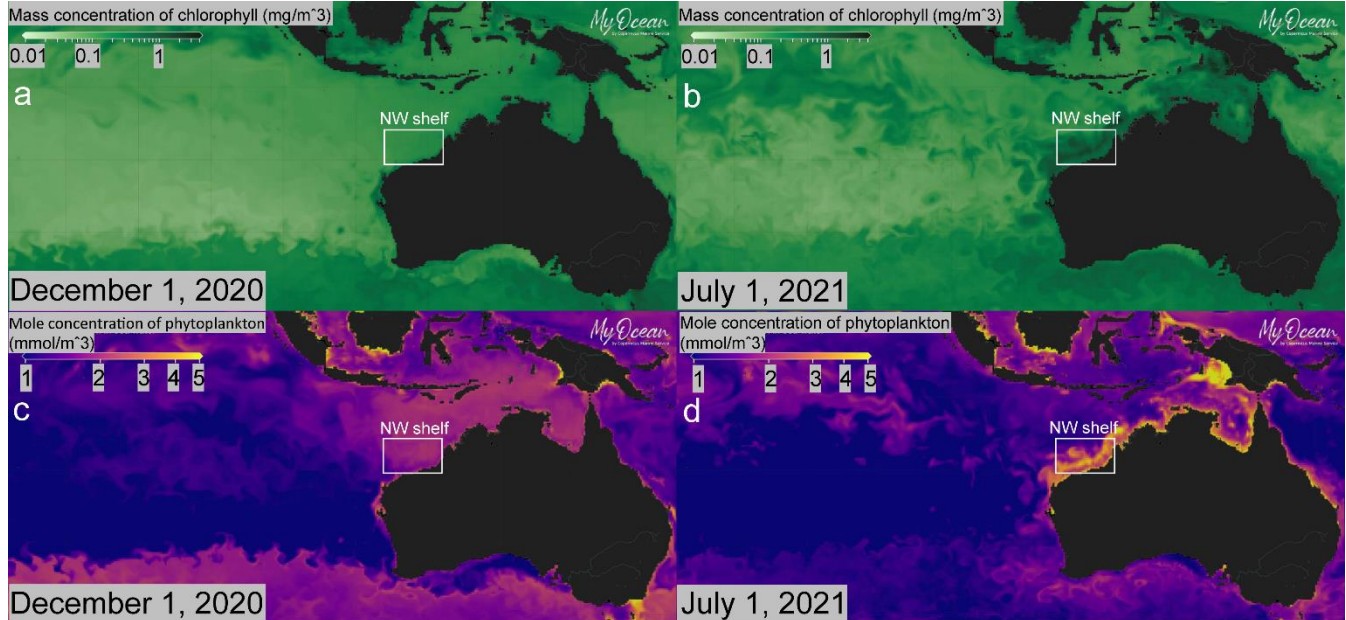

**Figure 3** Present-day ocean productivity maps of the southeastern Indian Ocean area for the 1st of December of 2020 (austral summer; **a, c**) and 1st of July of 2021 (austral winter; **b, d**). Reduced LC and increased water column stratification during austral summer leads to low concentrations of **(a)** chlorophyll and **(c)** phytoplankton in the NW Australian shelf area. Intense LC and storm track activity during austral winter leads to increased water column mixing in the NW Australian shelf and significantly higher **(b)** chlorophyll and **(d)** phytoplankton concentrations. Maps were generated using the ocean data visualization tools of E.U. Copernicus Marine Service Information (MYOCEAN PRO) and dataset layer GLOBAL_ANALYSIS_FORECAST_BIO_001_028 (https://doi.org/10.48670/moi-00015).

## 2 Material and Methods

### 2.1 Nannofossil Counts

IODP Expedition 356 Sites U1463 (145 m water depth; 18°59′S, 117°37′E) and U1464 (270 m water depth; 18°03.9115′S, 118°37.8935′E) (Figure 1) yielded a stratigraphic succession from the late Miocene to the early Pleistocene, with abundant, moderately preserved calcareous nannofossils. Note that the late Miocene to Pliocene sediments were deposited at much greater depths than the current water depths suggest, mirroring at the time of deposition middle/outer shelf to upper bathyal conditions (Christensen et al., 2017; Gallagher et al., 2017). Core sample ages are based on the tuned age models for Site U1463 (Groeneveld et al., 2021) and Site U1464 (Karatsolis et al., 2020).

A total of 209 samples (104 samples from Site U1463 and 105 samples from Site U1464) were prepared for micropaleontological analysis, with a sampling interval of ~1.5m (at least one sample per core section). Sample preparation followed the 'drop technique' (Bordiga et al., 2015). For each sample, 5 mg of dried bulk sediment was weighed and initially diluted with 20 mL of ammonia-buffered water. After short sonification (30-60 seconds), the suspension was passed through a 63 µm-meshed sieve and further diluted to three different final concentrations (bulk-weight equivalent concentrations of ~0,125 mg/mL; 0,08 mg/mL; 0,04 mg/mL). Subsequently, 1.5 mL of well-mixed suspension was placed on a cover slip with

a high-precision micropipette, and the sample was dried on a hotplate at 60°C. Finally, cover slips were mounted on glass slides with Norland Optical Adhesive (NOA61) and cured under a UV light source for approximately half an hour.

Micropaleontological analysis was conducted using polarized light microscopy at $1000 \times$ magnification, with at least 300 specimens counted and identified at genus level per sample. Nannofossils of the *Reticulofenestra* and *Gephyrocapsa* genera were further classified by size categories of <3μm (small), 3-5μm (medium) and >5μm (large). Samples from different dilution concentrations were selected, based on the abundance of nannofossils on the slides. Confidence intervals for nannofossil relative abundance were calculated using the binomial error function (95% CIs) in PAST 4 freeware (Hammer et al., 2001), as

suggested by Suchéras-Marx and others (2019). Subsequently, absolute abundances (in nannofossils per gram of sediment, N/g) of the different genera and morphospecies were calculated using the following equation (e.g., Koch and Young, 2007):

$$(1) \ \text{Abs. Abundance (AA)} = \frac{N \times A}{f \times n \times W} \ ,$$

where A is the area of the cover slip (mm$^2$), N is the total number of nannofossils counted, f is the area of one field of view (FOV; mm$^2$), n is the number of FOV, and W is the equivalent dry bulk sediment weight on the cover slip (g). Reproducibility

for the absolute abundance calculation is estimated to be ±10-15% (Bordiga et al., 2015).

Nannofossil fluxes were calculated following the formula:

$$(2) \ \text{NAR} = \text{AA} \times \text{LSR} \times \text{DBD} \ ,$$

where NAR stands for nannofossil accumulation rates (N/cm$^2$ kyr), AA is the absolute abundance of nannofossils (in N/g), LSR is the linear sedimentation rate (in cm/kyr) calculated between astronomically-tuned tie points and DBD is the average

dry bulk density (in g/cm$^3$), calculated for the studied interval from available data in the IODP LIMS online report portal (https://web.iodp.tamu.edu/LORE/). Nannofossil flux records older than 5.8 Ma at Site U1464 were not included in the interpretation of changes in nannofossil abundances, because they are biased by the very high sedimentation rates following the abrupt, late Miocene deepening of the Roebuck basin (Karatsolis et al., 2020). Nevertheless, relative nannofossil abundances, that reflect changes in the assemblages, extend the record at Site U1464 back to ~6 Ma.

**2.2 Correspondence analysis (CA) and Shannon diversity index (H)**

Correspondence analysis (CA) and calculation of the Shannon diversity index (H) (Shannon, 1948) were conducted using PAST 4 freeware (Hammer et al., 2001). We applied CA to the nannofossil time series data, in order to compare samples of different age based on the relative abundance of species present in each sample. This allowed for identification and visualization of possible breaking points where species composition and the structure of the assemblage significantly changed

through time. However, this ordination method does not provide measures of statistical significance, and therefore only serves to confirm, by observations in multivariate space, the main trends observed in the raw data. CA was chosen over canonical correspondence analysis (CCA), because our time series do not include any additional environmental parameters (no environmental gradient extraction). The H index was calculated to spot shifts in nannofossil diversity and identify intervals of

species overturn. The H index varies from 0 for assemblages consisting of a single species to high(er) values for assemblages with many species, but each represented by few individuals.

## 2.3 Nannofossil stratification index

A ratio between the three most common genera of the assemblage, namely bloom-forming *Reticulofenestra* and *Gephyrocapsa* species and *Sphenolithus* spp., was calculated to investigate relative changes in water column conditions and was named nannofossil stratification index (NSI). Species comprising this ratio are abundant throughout the record until the extinction of *Sphenolithus* spp. at ~3.54 Ma (Gradstein et al., 2012), while no long-term decreasing trend in the relative abundance of *Sphenolithus* was observed in the time preceding its last occurrence (LO; Figures A1, A2). The ratio was calculated as follows:

$$(3) \ \mathrm{NSI} = \frac{\mathrm{Sph}}{\mathrm{Sph} + (\mathrm{Rm} + \mathrm{Rs} + \mathrm{Gs})},$$

where Sph, Rm, Rs and Gs represent the relative abundances of *Sphenolithus* spp., medium-sized *Reticulofenestra* and small-sized *Reticulofenestra* and *Gephyrocapsa* species, respectively.

Of the nannofossil groups comprising this ratio, small sized *Reticulofenestra* species have been demonstrated to thrive in unstable conditions during the Pliocene, since they were favoured by more pronounced seasonality and tidal mixing (Auer et al., 2019; Ballegeer et al., 2012). With a similar ecological affinity, medium-sized *Reticulofenestra* species reflected neritic conditions with increased nutrient availability through local upwelling activity (Auer et al., 2019 and references therein). Small *Gephyrocapsa*, although also considered bloom-forming, highly opportunistic species, were adapted to warmer and more stratified water masses, as commonly inferred in subtropical continental margins (Auer et al., 2019; Boeckel and Baumann, 2008; Okada and Wells, 1997; Takahashi and Okada, 2000). *Sphenolithus* spp. on the other hand, although of complex paleoecological affinity that ranges from low-latitude, oligotrophic warm waters (e.g., Gibbs et al., 2004; Haq, 1980; Haq and Lohmann, 1976) to restricted mesotrophic ocean masses (Wade and Bown, 2006), has been commonly grouped with *Discoaster* as characteristic of warmer, more stratified water conditions in contrast to the bloom-forming *Reticulofenestra* and *Gephyrocapsa* species (e.g. Gibbs et al., 2005; Haq and Lohmann, 1976). Similar ratios between "cool", mesotrophic species and "warmer", oligotrophic species have been used in assessing the paleoecology during other geological periods. For example, the ratio between species of the genus *Toweius* with *Sphenolithus* and *Discoaster* has been used as a paleoenvironmental index for the Paleocene-Eocene thermal maximum (Gibbs et al., 2010). Species of the genus *Toweius* have been characterised as meso/eutrophic, having occupied a niche similar to the modern bloom-forming species (e.g. Bralower, 2002; Gibbs et al., 2010). We therefore propose that the NSI can serve as a useful indication of changes in water column mixing and nutrient availability and has the potential to record relative stratification above the NW Australian shelf during the latest Miocene to Pliocene.

## 2.4 Shelf-to-offshore paleotemperature gradients

Paleotemperature gradients between sites located in the eastern Indian Ocean (DSDP Site 214 and ODP Site 763) and the NW Australian shelf (IODP Site U1461) (Figure 1), were calculated based on available proxy-based estimates. At Site U1461, two paleotemperature proxy records include archaeal membrane lipids ('GDGTs')-based $TEX_{86}$ and alkenone-based $U_{37}^{k'}$ temperatures (He et al., 2021), spanning the interval from ~6-3.5 Ma. At Site 763 and Site 214, available records consist of Mg/Ca-derived temperatures from *Trilobatus sacculifer* foraminifera (Karas et al., 2009, 2011). A direct site-to-site comparison based on the same paleotemperature proxies is therefore not possible. However, the quality and temporal resolution of each record are sufficient for the purpose of contrasting overall and long-term shelf-to-offshore temperature trends, despite potential biases between proxies. Alkenone-producing haptophytes live in the photic zone (e.g. Eglington et al., 2001; Müller et al., 1998) and *Trilobatus sacculifer* calcifies in the upper 50 m of the water column in the tropics (e.g., Anand et al., 2003; Dekens et al., 2002), therefore the $U_{37}^{k'}$ and Mg/Ca proxies reliably reflect sea surface temperature (SST). On the other hand, $TEX_{86}$ estimates appear to be more complicated and although widely used to reconstruct SSTs, they may in some cases record temperatures down to >200 m depth (van der Weijst et al., 2022). This is mainly attributed to the fact that GDGTs are produced by Thaumarchaeota throughout the water column and therefore deeper-water species also contribute to the sedimentary composition of this proxy (Kim et al., 2016). Along the Australian shelf, Pliocene and Pleistocene records of $TEX_{86}$ have been used both as SST indicators (e.g., De Vleeschouwer et al., 2019; Smith et al., 2020), or containing a deeper signal (Smith et al., 2013), with the record from Site U461 assumed to reflect an integrated 0-200m water column depth (He et al., 2021). However, during the early Pliocene, only a small vertical thermal difference was inferred at this site, based on the difference between the "shallower" $U_{37}^{k'}$ and the "integrated 0-200m" $TEX_{86}$ temperature signals (He et al., 2021). Because of this observation, as well as the lack of Mg/Ca records and no $U_{37}^{k'}$ data older than 5.5 Ma on the shelf, we included $TEX_{86}$ in the comparisons. Four pairs of temperature gradients were calculated between samples with the closest assigned age. The gradients between $U_{37}^{k'}$ proxy estimates and Mg/Ca-based estimates were labelled as "ΔSST" ($\Delta SST_{U1461-763}$, $\Delta SST_{U1461-214}$), whereas those between $TEX_{86}$ and Mg/Ca as "ΔT" ($\Delta T_{U1461-763}$, $\Delta T_{U1461-214}$), to account for a potential bias from a subsurface signal in the $TEX_{86}$ proxy and track related discrepancies. The median of the age difference between sample pairs was calculated as an error in our considerations of long-term trends.

## 3 Results

### 3.1 Nannofossil relative abundances and fluxes

Calcareous nannofossils are very abundant and moderately preserved throughout the studied interval. Preservation was initially assessed to be moderate to good (Gallagher et al., 2017b, Gallagher et al., 2017c), but a re-assessment through visual inspection showed that good preservation was actually rare. Coccolith NAR and relative abundances reveal highly similar patterns at both sites, with bloom-forming *Reticulofenestra* (<5μm) and *Gephyrocapsa* (<3μm) species always dominating the assemblage

(combined ~60-97%; Figures A1, A2). The smallest bloom-forming species (<3μm *Reticulofenestra* and *Gephyrocapsa*) were common to abundant throughout, ranging from ~30-97%. The next most common species was medium-sized *Reticulofenestra* (3-5μm), which constituted ~2-47% of the assemblage at Site U1463 and ~6-52% at Site U1464. *Sphenolithus* spp. reached up to ~20% at Site U1463 and ~23% at Site U1464, until it dramatically decreased at ~3.54 Ma, following the global pattern of extinction of the genus (Figures A1, A2). Other commonly occurring species were *Calcidiscus* spp., that constituted up to ~7% at both sites, *Umbilicosphaera* spp., with relative abundances up to ~10% at Site U1463 and ~9% at Site U1464 and larger (>5μm) *Reticulofenestra* species, with percentages up to ~11% at Site U1463 (Figure A3) and ~9% at Site U1464 (Figure A4). Consistently present species, but with very low relative abundances, were *Helicosphaera* spp. and *Discoaster* spp. Total NAR ranged from ~$2\times 10^{10}$ to ~$3\times 10^{11}$ N/cm$^2$ kyr at Site U1463 and ~$2\times 10^{10}$ to ~$2\times 10^{11}$ N/cm$^2$ kyr at Site U1464 (not shown). Two time intervals can be identified that represent distinct changes in nannofossil assemblage composition and NAR of the most common species. The first occurred from 5.4–5.2 Ma and was characterized by a ~10% increase in relative abundance (and NAR) of bloom-forming species and a synchronous ~10% decrease in relative abundance (and NAR) of *Sphenolithus* spp. at both sites. The fact that the burial fluxes (NAR) changed in opposite direction supports the assumption that these taxonomic groups have distinct ecologies and were both affected simultaneously (Figures A5, A6). At the same time, bulk sediment mass accumulation rates (MARs) do not demonstrate any notable, sustained change, indicating that the observed pattern in the assemblage composition was driven by changes in species' abundance (Figures A5, A6). *Umbilicosphaera* spp. and large *Reticulofenestra* species had a distinctive low in relative abundances during this period, whereas *Calcidiscus* spp. was not significantly influenced (Figures A3, A4).

The second interval occurred from 4.6–4.4 and represents a rapid decrease in total NAR, mainly driven by the decrease in abundance of small *Reticulofenestra* species (Karatsolis et al., 2020), as well as a basin-wide decrease in low-latitude paleoproductivity (PP; Karatsolis et al., 2022). During this interval, the relative abundances of other common species such as *Calcidiscus* spp. and *Umbilicosphaera* spp. increased and remained steadily higher compared to the interval before 4.6 Ma (Figures A3, A4), while their fluxes were not significantly affected (Figures A7, A8). Large *Reticulofenestra* species virtually disappeared during this interval (Figures A7, A8), following the rapid decrease in NAR of the smaller species of the same genus (Figures A5, A6). As NARs of small *Reticulofenestra* spp. decreased, the small *Gephyrocapsa* spp. started to increase (~4.45–4.42 Ma) in percentage. Small *Gephyrocapsa* spp. became dominant from ~4.2 Ma until the end of the studied interval, with maximum NAR values of ~$1 \times 10^{11}$ N/cm$^2$ kyr at Site U1463 and ~$5 \times 10^{10}$ N/ cm$^2$ kyr at Site U1464, which corresponds to ~70% and ~60% of the total assemblage respectively. After ~4.4 Ma, the NAR of *Sphenolithus* spp. returned to higher values, especially at Site U1463, showing distinct peaks that are centred at ~4.1 Ma and 3.7 Ma. Its relative abundance increased too, occasionally reaching values of >10%, although it never reached maxima as during the latest Miocene (Figures A1, A2).

 **3.2 Nannofossil multivariate analysis**

Changes in NSI covary throughout the studied interval and correlate well between the two sites (Figure 4a). A stepwise decrease is observed from 5.4–5.2 Ma, which mirrors the overall decrease in relative abundance and NAR of *Sphenolithus* spp. and the synchronous increase in small *Reticulofenestra* species. After ~4.2 Ma, NSI shows higher variation with short intervals of higher values (4.2-4 Ma and 3.8-3.54 Ma). These intervals correspond to peaks in relative abundance of *Sphenolithus* spp.

Between 4 and 3.8 Ma, NSI shows low values, whereas after 3.54 Ma, the LO of *Sphenolithus* spp. makes the index non-applicable. The H-index (Figure 4b) demonstrates that the most distinct change, which collectively influenced the nannofossil species distribution, occurred from 4.6–4.4 Ma and was marked by a stepwise increase in diversity. Notably, no sustained change in diversity is observed during the interval of stepwise decrease in NSI (5.4–5.2 Ma; Figure 4a).

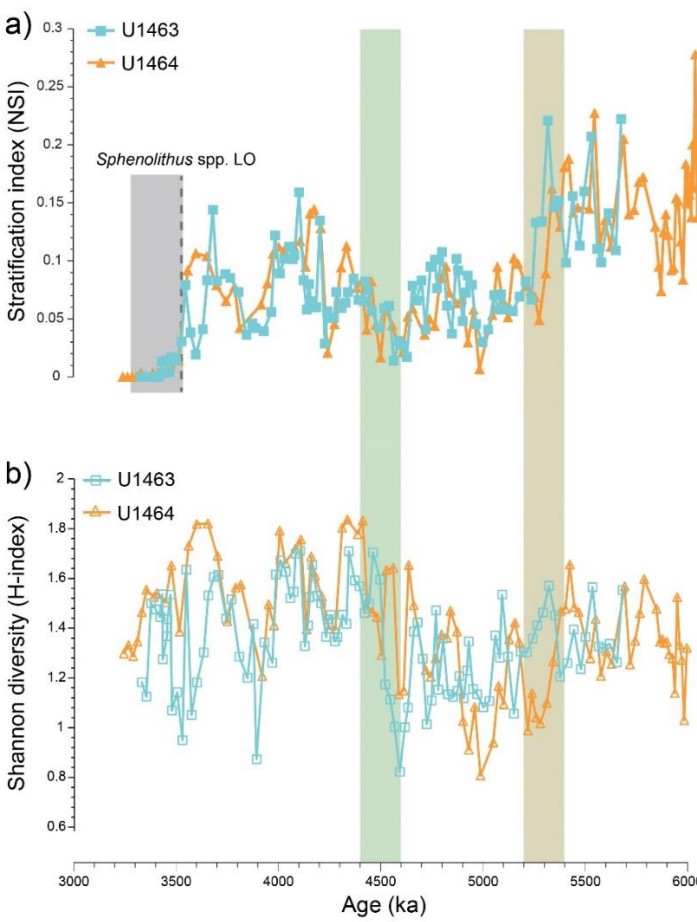


**Figure 4 (a)** Miocene to early Pliocene nannofossil stratification index (NSI) at IODP Sites U1463 (blue squares) and U1464 (orange triangles). Dashed grey line indicates the last occurrence (LO) of *Sphenolithus* spp. during the early Pliocene (~3.54 Ma). Therefore, this NSI cannot be used for younger samples (grey shading). **(b)** Shannon diversity (H-index) at the same sites. Brown shading highlights the stepwise decrease in NSI and increase in NAR of bloom-forming species. Green shading represents the interval of significant decreases in
paleoproductivity (PP) and NAR of bloom-forming species.

By color-coding the samples based on their age across the main time intervals of interest (5.4-5.2 Ma and after 4.6–4.4 Ma; Figure 5), we test if any of the observed changes in the abundance of dominant species were accompanied by shifts in the relationship between relative abundances of species in multivariate space. Correspondence analysis at both sites mainly highlights the changes that occurred during 4.6–4.4 Ma, almost splitting the samples that belong to the period before the rapid decrease in NAR from those after that across the primary CA axis (Figure 5a, b). It also groups the species that were more dominant before and after ~4.4 Ma with the respective interval of dominance (low angle from the (0,0) source between species and sample age). Species such as small *Reticulofenestra* and *Sphenolithus* spp. are separated from small *Gephyrocapsa*, *Calcidiscus* spp. and *Umbilicosphaera* spp. (Figure 5). Medium-sized *Reticulofenestra* appears to have a less clear affinity, being close to the primary axis and on opposite sides of it at Sites U1463 and U1464. This can probably be attributed to the fact that this species showed relatively stable relative abundances and two distinct peaks before and after the 4.6–4.4 Ma event (Figure A9), which could also explain the few before/after 4.4 Ma samples that fall on the same side of the axis (Figure 5a, b). Interestingly, species loadings and <4.4 Ma sample scores correlate well, and both appear to be situated further from the (0,0) source, demonstrating that the association of common species with the younger part of the record is stronger.

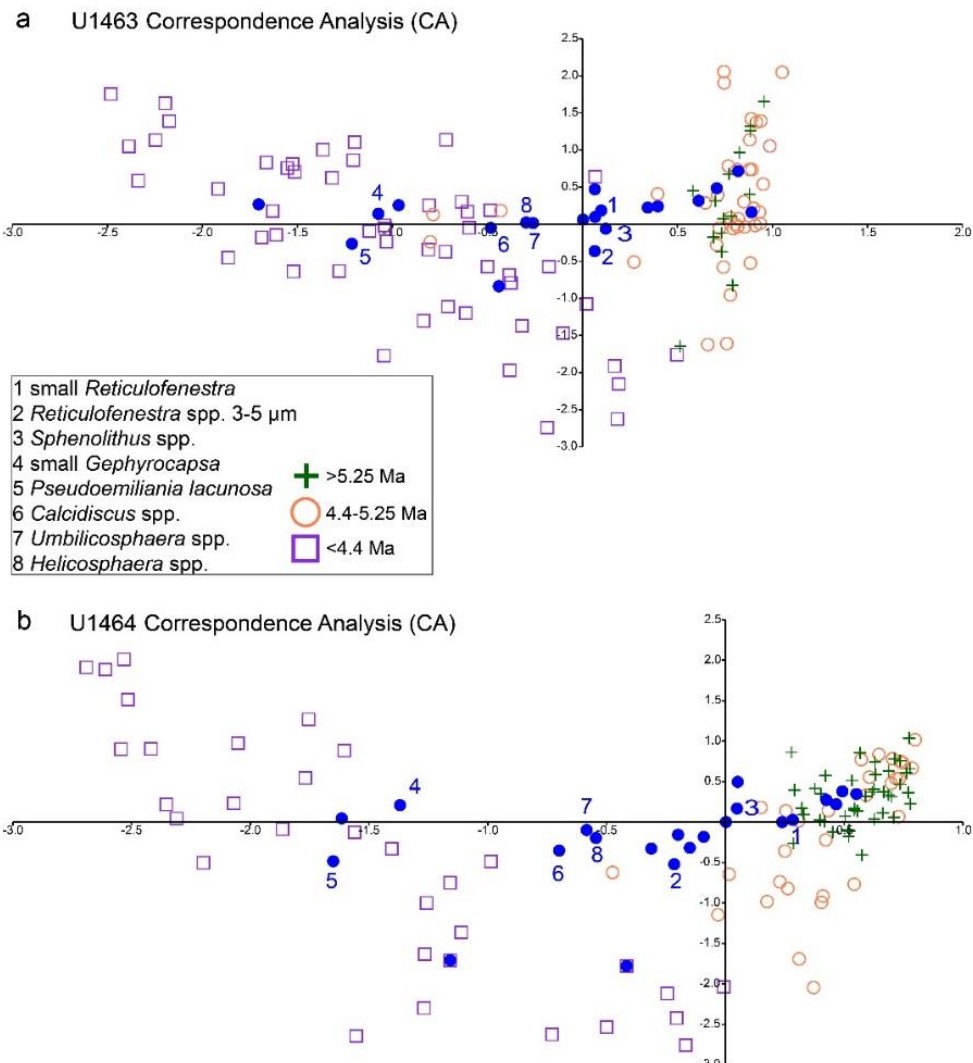

**Figure 5** Correspondence analysis for IODP Sites U1463 **(a)** and U1464 **(b)**. Orange circles represent samples older than 5.25 Ma and together with the green plus signs (4.4-5.25 Ma) constitute the records prior to the significant decrease in NAR that occurred at 4.6–4.4 Ma. Purple squares represent sample ages <4.4 Ma. Numbered (1-8) blue dots represent the most common nannofossil species (see legend).


### 3.3 Paleotemperature gradients

The median difference between closest sample ages for any pair of sites ranged between ~2-10 kyr (Table A1), although in one case, due to a gap of >200 kyr in sample spacing at Site 214, the sample age difference was ~100 kyr (~3.75-3.96 Ma) for
gradients involving this site. However, this gap in paleotemperature estimates occurs towards the end of our studied interval and therefore does not limit investigating possible links to sustained changes in the nannofossil assemblages across the Miocene-Pliocene boundary and between 4.6–4.4 Ma. Despite the different previous interpretations of the temperature proxies, their gradients demonstrate highly comparable long-term patterns; they co-vary throughout the Pliocene and can be separated in three distinct intervals (Figure 6). Before ~5.2 Ma, all gradients have positive values, demonstrating that temperatures at
Site U1461 were consistently higher than those in the eastern Indian Ocean. However, only one gradient ($\Delta T_{U1461-763}$) is based on more than 3 samples and is the most informative gradient for the interval >5.2 Ma and the latest Miocene. Between 6-5.2 Ma, this temperature gradient had a median value of ~2 °C and decreased to -0.6 °C for the interval between 5.2-2.5 Ma, likely driven by a cooling at Site U1461 and a warming at Site 763 (Figure 6). This indicates that $TEX_{86}$ temperatures at Site U1461 transitioned from a phase of steadily warmer conditions to slightly cooler values compared to Site 763. In general, $\Delta SST$ values
are smaller than $\Delta T$, with median values of -0.2 and -0.5 °C ($\Delta SST_{U1461-763}$, $\Delta SST_{U1461-214}$) *versus* -0.63 and -1.14 °C ($\Delta T_{U1461-763}$, $\Delta T_{U1461-214}$) for the interval from 5.2-3.5 Ma (see Table A1). Median values also reveal that, both for the $\Delta SST$ and the $\Delta T$ records, gradients are generally greater between Site U1461 and Site 214 than those between Site U1461 and Site 763. After 5.2 Ma and until ~4.3 Ma, gradients have negative values (reflecting lower temperatures over the shelf) and show small fluctuations between 0 and -2 °C, whereas after ~4.3 Ma fluctuations become more intense and vary from ~ -3.5 to 2 °C.
Despite this strong variation, positive gradient values never exceed those observed before 5.2 Ma and remain mostly negative, indicating that warmer conditions were prevailing in the eastern Indian Ocean, whereas waters over the shelf were generally cooler.

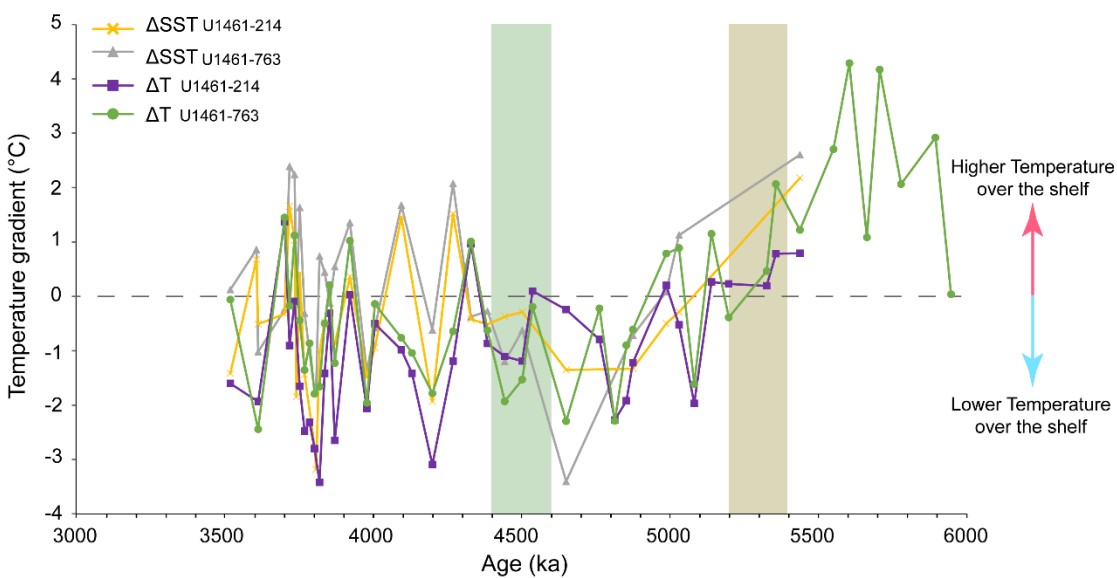

**Figure 6** Temperature gradients (ΔSST and ΔT) between IODP Site U1461 (NW Australian shelf), ODP Site 763 (Exmouth Plateau) and DSDP Site 214 (eastern Indian Ocean). Gradients are calculated between $U_{37}^{k'}$ and Mg/Ca records (ΔSST) and the $TEX_{86}$ and Mg/Ca records (ΔT) for the interval 6-3.5 Ma. Horizontal grey dashed line indicates no difference in temperature (ΔSST, ΔT = 0°C). Brown shading corresponds to the stepwise decrease in NSI and increase in NAR of bloom-forming species. Green shading represents the interval of significant decrease in PP and NAR of bloom-forming species.

## 4 Discussion

### 4.1 Water column mixing and nutrient availability on the NW Australian shelf

The common species comprising the NSI, namely small bloom-forming species and *Sphenolithus* spp., provided the bulk of the nannoplankton biomass and productivity and can therefore be expected to represent species that had broad ecological tolerances and the potential to record climate-biota interactions and feedbacks over long time intervals (e.g., Hannisdal et al., 2012; Henderiks et al., 2020). Because the nannofossil abundance in each sample represents an average, smoothed record over thousands of years, it would be impossible to directly infer changes in seasonal and interannual variation of the ITF and LC in a similar way that modern plankton observations and climate models do. Nevertheless, it can be expected that changes in ocean circulation or seasonality that significantly influenced the flow path of major boundary currents over the course of thousands of years, would be mirrored in the mean nannofossil assemblage, favouring taxa that benefited seasonally or interannually by the prevailing conditions. Therefore, the NSI offers an indication of persistent overall changes in water column stratification and mixing, that could be driven by a long-term switch in seasonal dynamics, ITF flow or LC intensity during the latest Miocene to Pliocene.

On this basis, the NSI reveals a stepwise phase of reduction in stratification and intensified nutrient replenishment in the upper water column, that took place around the Miocene-Pliocene boundary (5.4-5.2 Ma; Figure 4a). This suggests significant environmental changes during that time and a transition from a more tropical stratified environment to a modern-like dynamic

shelf area, favouring bloom-forming species over *Sphenolithus* spp. After its reduction, NSI never returned to its late Miocene levels, demonstrating that increased water column mixing was a prevailing regime in the area from 5.2-3.54 Ma, although an interval of overall higher NSI, with stronger amplitude variations between higher and lower phases can be observed from ~4.2 to 3.54 Ma (Figure 4a). This indicates that intervals of relatively stronger stratification occurred between 4.2-4.0 Ma and 3.8-3.54 Ma (higher NSI phases), while water column mixing was more intense between 4-3.8 Ma.

Three possible mechanisms related to regional paleoclimate and LC dynamics could explain the observed pattern. The first would suggest a strong effect of changes in upper water column stratification on primary productivity due to a LC acceleration, as it has been previously documented in a series of modern observational data and Pleistocene paleoclimatic records. Specifically, intensified LC activity has been linked to increased eddy formation that promotes productivity across the western Australian continental shelf (Auer et al., 2021; Feng et al., 2009; Furnas, 2007; Koslow et al., 2008; Thompson et al., 2011).

This mechanism has also been proposed for other coastal areas influenced by boundary currents such as the southwestern Australian shelf (Harris et al., 1987; Koslow et al., 2008) and the Japanese coast (Kimura et al., 2000). Additionally, the modern observational and simulated data also show that the NW Australian shelf demonstrates increased water column mixing and phytoplankton productivity close to the shelf during austral winter when LC is stronger (Figures 2 and 3). Since the presence of the LC depends on steeper steric height gradients between the source areas and the southern latitudes, we

hypothesize that such steepening intensified during the early Pliocene, triggering enhanced seasonal flow of this major boundary current. In turn, intensified eddy activity and vertical mixing during the winter season and a subsequent "pumping" of nutrients from the continental slope onto the outer continental shelf, generated the observed long-term patterns in NSI, as well as the increase in NAR of bloom-forming species (Figures A5, A6).

A second mechanism that could explain the NSI decrease, is the overall increase in convective mixing across the continental

shelf area due to intensified cooling in the upper water column, as well as intensified storm activity during the winter period. These processes are known to have the potential for generating blooms by mixing the water column and bringing nutrients to the euphotic zone (e.g Chen et al., 2020; Longhurst, 2001). This mechanism was proposed for modern winter blooms along the west coast of Australia, which have been attributed to mixing of the upper water column and a shoaling of the nutricline and chlorophyll maximum layer (Koslow et al., 2008).

The third mechanism comes in contrast to the first one and is based merely on the physical properties of the LC. According to that scenario, the warm SSTs and nutrient deficient waters that characterize the LC (Godfrey et al., 1986; Ridgway and Condie, 2004) should inhibit upwelling activity and stratify the water column, and therefore lead to overall decreased primary productivity, even across the continental shelf. Specifically, this relationship has been used to support the plausible LC activity in the NW Australian shelf from 6-4 Ma, based on the low concentrations in the sediment of isoGDGTs and C28–C30 sterols

(He et al., 2021). Similarly, a cooling during the Pleistocene along the NW Australian shelf has been previously attributed to a slowing down of the LC that would have allowed for intensified regional upwelling (Smith et al., 2020). Following the latter scenario, the reduction in NSI that we observe would be the result of a LC weakening, that allowed for seasonal upwelling and nutrient enrichment in the upper water column. However, it is worth noting that a similar cooling and intensification of

upwelling could be achieved by a long-term strengthening of the summer north-westerly winds that overcome the south-
easterly trade winds and have the potential to generate seasonal upwelling. Since the above-mentioned scenarios are not
mutually exclusive and involve processes that could have co-existed over long periods of time, a combination of mechanisms
could explain the observed changes in NSI.

### 4.2 Paleotemperature and inferred ocean circulation patterns

Temperature gradients between the eastern Indian Ocean and the NW Australian shelf sites further support significant changes
in ocean circulation during the late Miocene to Pliocene and compliment the observed changes in the nannofossil assemblages.
In general, the higher $\Delta T$ gradient (in absolute values) after 5.2 Ma compared to the $\Delta SST$, supports the hypothesis that $TEX_{86}$
temperatures were influenced by a slightly deeper and cooler signal of the integrated upper 200 m of the water column (He et
al., 2021; Smith et al., 2013). Although these differences are consistent throughout the record, they are generally small and
therefore in good accordance with the low gradient between the integrated water column temperature and the SST ($\Delta T_{TEX86-}$
$_{U^k_{37}}$) for the interval between 6-3 Ma at Site U1461 (median value of ~1.1°C; He et al., 2021). More importantly, the temperature
differences derived from all proxies between the NW Australian shelf and the eastern Indian Ocean site show similar variation
and record a stepwise decrease of the temperature gradient between the two areas, driven by a cooling in the former area and
a warming in the latter (Figure 7a, b). The overall change can be mainly tracked by the median values of $\Delta T_{U1461-763}$ before and
after 5.2 Ma, which suggest a shift of ~2.5°C that closes the temperature gap between the shelf and the eastern Indian Ocean,
while flipping the temperature relationship between the two regions by making the latter one warmer. Similar temperatures
across the NW Australian shelf and the eastern Indian Ocean indicate well-established ITF connectivity, in good agreement
with the suggested warm and humid conditions that were prevailing in the area since the late Miocene (Christensen et al., 2017;
Karatsolis et al., 2020; Stuut et al., 2020). Interestingly, this change also coincides with the decrease in NSI from 5.4–5.2 Ma,
suggesting a possible link between the changes in temperature and the stratification index (Figure 7 a, b), either by a decreased
tolerance of *Sphenolithus* spp. to relatively cooler temperatures over the shelf, or through changes in nutrient availability due
to intensified seasonal mixing and/or upwelling that favoured opportunistic species. On a similar note, both the NSI and the
paleotemperature gradient records increase in amplitude between ~4.2-3.54 Ma, showing intervals of higher and lower values
and peaks that are well correlated (both centred at ~4.1 and 3.7 Ma). That could indicate a similar link of temperature and NSI
on shorter timescales, although such a correlation would need higher resolution paleotemperature and nannofossil records in
order to be sufficiently established.

In principle, a cooling at Site U1461 could be explained by (i) cooler surface water that began to influence the area and
decreased the average temperature, (ii) the weakening of the ITF and LC which would have allowed for seasonal upwelling of
deeper, colder water to occur, or by (iii) a sustained increase in vertical mixing, that led to an overall decrease in temperature
in the integrated upper water column. However, since evidence for significant restriction of the ITF or a switch of LC source
are suggested to have occurred much later, towards the late Pliocene (e.g. Christensen et al., 2017; Karas et al., 2011; De
Vleeschouwer et al., 2019), we suggest that an increase in water column mixing is the most probable scenario to have generated

the observed changes in the average temperature signal. Furthermore, if this increase in mixing was driven by an overall intensification of seasonal extremes, it could have resulted in a deeper mixed layer and increased eddy formation along the Australian shelf because of both increased storm track and LC activity. Further evidence for the increased presence of the LC

in the area comes from the temperature changes at ODP Site 763. Records from this site have been previously used to infer the presence of this boundary current in the geological past (e.g., Auer et al., 2019; Karas et al., 2011) and it is located in an area that has been commonly used to test LC activity through Ocean General Circulation models (Godfrey, 1996; Hirst and Godfrey, 1993). At Site 763, we observe a distinct increase in SST of ~2°C (Figure 7b), a shift that would not have been possible if the source of the LC had become cooler, or if the LC was weakened in order to accommodate the hypothesis of cooling in the

shelf area. At the same time, no warming is observed further offshore at DSDP Site 214, strengthening the hypothesis that Site 763 tracks the more regional effect of the LC.

Combined, the changes in NSI and paleotemperatures could be explained by an overall long-term increase in water column mixing over the shelf, which could have been driven by an intensification of the atmospheric pressure and steric height gradients over NW Australia. Specifically, such intensification would translate in stronger trade winds and LC intensification,

a hypothesis that is supported by the observed warming at Site 763. Meanwhile, the decrease in NSI could have been generated either by a sustained increase in vertical mixing and eddy formation over the shelf during the winter season, or by the stronger reversal of the trade winds and monsoonal north-westerlies during the summer, which can promote regional upwelling. Both scenarios, which fall close to a combination of the first and second mechanisms presented in section 4.1, could have been the result of an invigoration in seasonal extremes that marked the transition to a less tropical, less stratified environment and a

significant shift in the paleoecological and paleoceanographic regime in the area (Regime 1 (more stratified, weaker LC) to Regime 2 (less stratified, stronger LC); Figure 7). The reverse scenario would apply for the periods before 5.4 Ma and (less intensely) after ~4.2 Ma, when higher NSI and the dominance of *Gephyrocapsa* spp., a bloom-forming group of species with increased tolerance of warmer and more stratified conditions, is observed. Further offshore, at Site 763, where the effect of continental shelf eddy formation and seasonal upwelling is weaker, the invigorated presence of the LC after ~5.4–5.2 Ma, is

solely translated by an overall increase in SSTs.

Although the records investigated in this study do not allow for an extended evaluation of more detailed changes in seasonal dynamics and the driving mechanisms behind this regime shift, the globally elevated temperatures that marked the beginning of the Pliocene (~5.3 Ma; Fedorov et al., 2006) could have played a role in increasing monsoonal precipitation and tropical cyclones (Fedorov et al., 2010), therefore invigorating water column mixing in low latitudes. However, this warming was

suggested to be an expression of a permanent El Niño state (Fedorov et al., 2006; Wara et al., 2005), which would contradict our argument of an intensified seasonal presence of the LC, since this current is expected to be significantly weaker under such conditions (e.g., Feng et al., 2003, 2009). When compared to other studies, our interpretation falls close to the hypothesis by He and others (2021), namely that the LC was active during the early Pliocene and from 6-4 Ma, before it weakened after ~4 Ma. On the other hand, De Vleeschouwer and others (2022) argued for a strong LC from ~4-3.7 Ma and a weaker phase from

~3.7-3.1 Ma, a reconstruction we cannot evaluate using the NSI because this index cannot be applied after 3.54 Ma (see Figure

4a). More importantly, our study indicates the complex, differential effects that sustained changes in wind and ocean circulation patterns in major currents such as the LC, would have had on the shelf and further offshore.

## 4.3 Broader-scale changes in paleoproductivity and paleoecology

Apart from regional ocean circulation changes, nannofossil assemblages recorded on the NW Australian shelf may reflect broader-scale or even global patterns in plankton evolution and variations in the ocean's nutrient budget. The studied time interval falls within a period of elevated biogenic sedimentation called the late Miocene to early Pliocene biogenic bloom (e.g., Dickens and Owen, 1999; Diester-Haass et al., 2005; Farrell et al., 1995; Hermoyian and Owen, 2001), which has been observed in numerous locations in all major oceans and potentially came to an end, at least in low latitudes, with a sharp decrease in PP that occurred between 4.6–4.4 Ma (Karatsolis et al., 2022). This global event was also expressed at Sites U1463 and U1464 by an abrupt decrease in nannofossil fluxes, as well as a significant restructuring of the nannofossil assemblages, as indicated by the CA (Figure 5) and the increase in H-index (Figure 4b). Despite the relative abundance changes, small and medium-sized, bloom-forming calcareous nannoplankton dominated the assemblage throughout the studied interval. This could partially be explained by the constant supply of nutrients because of a steady terrigenous runoff (Karatsolis et al., 2020) onto the outer shelf/upper bathyal paleoenvironment, where the studied sites were situated during the "Humid Interval" (Christensen et al., 2017; Gallagher et al., 2017). However, it is also in good agreement with a basin-wide eutrophication event, that spanned from ~8.8 Ma (Imai et al., 2015) to ~3.5 Ma; during which *Reticulofenestra haqii* (medium-sized) and *Reticulofenestra minuta* (small-sized) assemblages dominated in the Indian Ocean (Young, 1990). Within the same period, a *Gephyrocapsa* dominance interval (acme) has been recorded from 4-3 Ma (Young, 1990), consistent with the observed progressive dominance of this species over the NW Australian shelf that followed the termination of the biogenic bloom. However, we can now confirm that a complete dominance overturn was never achieved, as evident by the stepwise increase in diversity at 4.6–4.4 Ma (H-index; Figure 4b), which was partly supported by the increase in relative abundances (but not fluxes) of *Calcidiscus* spp. and *Umbilicosphaera* spp. (Figures A3, A4, A7, A8). The latter taxa were probably affected less by a basin-wide decrease in nutrient availability and mainly mirrored the regional warm conditions over the shelf. At roughly the same time, large *Reticulofenestra* species virtually disappeared. This falls close to the timing (~5 Ma) of a distinct decrease in the modal size of *Reticulofenestra* species that has been recorded both in the eastern Indian Ocean (ODP Site 757) and the NW Pacific (ODP Site 1201) by Imai et al. (2020), although these authors suggested that the event was restricted to open ocean sites and therefore absent in more coastal locations near the shelf. A trend towards smaller nannofossil sizes continued during the Pliocene (~4–3.8 Ma) with evolutionary events such as the first occurrence (FO) of *Pseudoemiliania* spp. (Figure A10) and the LO of *Reticulofenestra pseudoumbilicus* (Aubry, 2007).

Interestingly, although significant assemblage changes occurred during the PP decrease, there was no major change in NSI (Figure 7). This supports the hypothesis that a broader decrease in nutrient availability occurred from 4.6–4.4 Ma, one that affected the abundance of dominant species as a whole, but not the relative ecological success between common taxonomic groups (bloom-forming versus *Sphenolithus*). The fact that the NSI remained unaltered underpins the potential of this index in

decoupling long-term changes in relative dominance of species, controlled by regional ocean circulation patterns and seasonal water column mixing intensity, from large-scale processes influencing PP through changes in the total ocean nutrient budget. Further evidence for a decrease in fluxes or significant changes in the nannofossil assemblages, that would have marked the end of the biogenic bloom, are still limited for the rest of the Indian Ocean. Some observations of nannofossil export production from the open ocean ODP Site 757 (Figure 1) support a reduction in burial fluxes in good accordance with our results (Imai et al., 2020). Another study that investigated the nannofossil assemblages at ODP Site 762 (Figure 1), which is located closer to the shelf, recorded a significant decrease in nannofossil fluxes between ~4.2-3.9 Ma, with the nannofossil fluxes returning to higher values after that (Imai et al., 2015). However, it is important to note that the age models of these studies were based on few early Pliocene biostratigraphic tie-points, in contrast to the orbitally-tuned records presented herein. Therefore, the exact timing of events within <0.5 Ma precision should be considered with caution.

As far as the dominance of small *Gephyrocapsa* is concerned, several acme events of this species have been previously reported during the Pliocene and Pleistocene, but they seem to have been largely diachronous between ocean basins (Auer et al., 2019; Ballegeer et al., 2012; Gibbs et al., 2004; Marino and Flores, 2002). This supports the existence of several controlling mechanisms for its dominance, age-model discrepancies, or a non-linear effect of a long-term ecological pressure across basins. Despite the diachronicity, it appears that small *Gephyrocapsa* had an opportunistic advantage in filling the available niches, and a paleoecological advantage to dominate the assemblage in several different paleoenvironments during the early Pliocene. Its increase in relative abundance can already be observed on the NW Australian shelf during, and directly after, the significant reduction in NAR of the opportunistic small *Reticulofenestra* species (4.6–4.4 Ma; Figures A5, A6). However, small *Gephyrocapsa* clearly dominates after 4.2 Ma, and this could be related to our hypothesis of a relative increase in stratification and reduced seasonal mixing of the water column during that time. The subsequent widespread acme in the Indian Ocean (Young, 1990) could have been a result of overall, basin-wide warming and more stratified conditions, a hypothesis that is supported by the very high early Pliocene temperatures in both open ocean Indian Ocean sites (Sites 214 and 763; Karas et al., 2011) from ~4.3–3.5 Ma. Nevertheless, additional high-resolution nannofossil records are needed to confirm the diachronicity between small *Gephyrocapsa* acme events and the shifts in the nannofossil assemblages that marked the termination of the biogenic bloom.

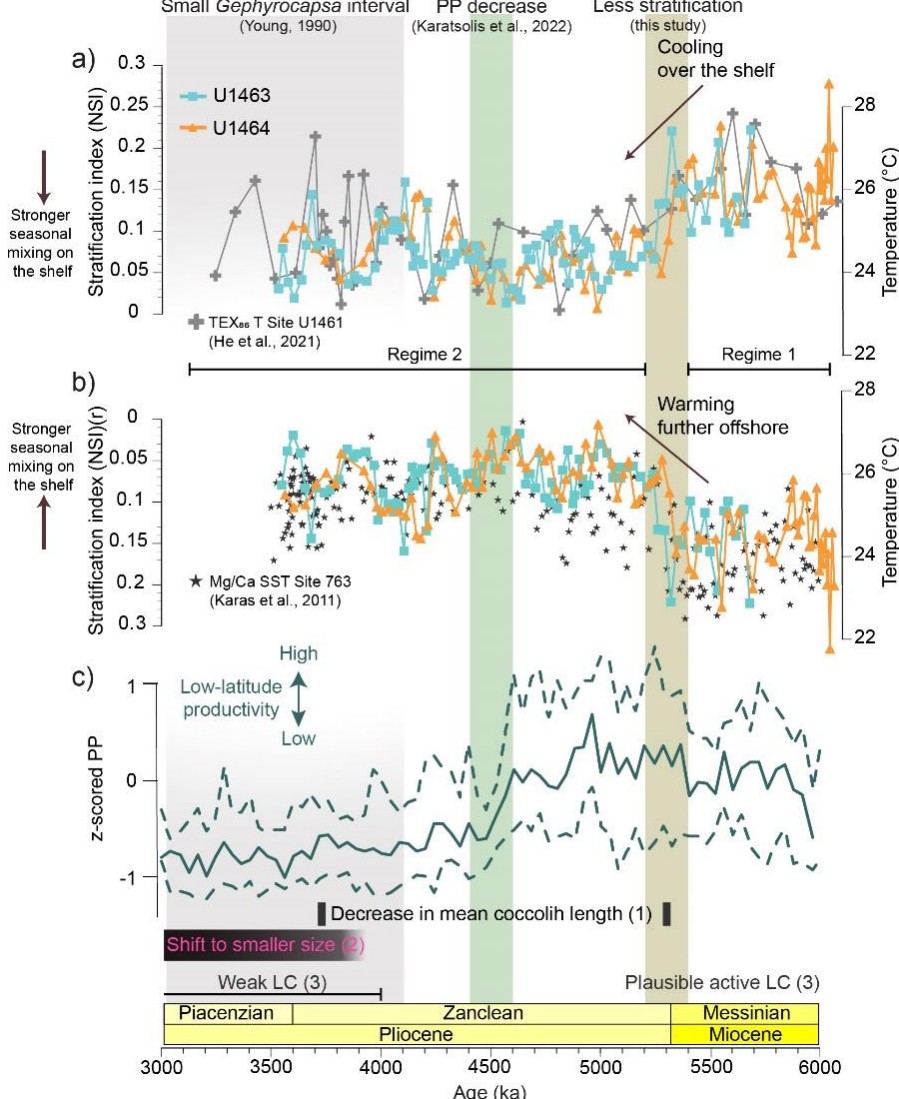

**Figure 7** Correlation between paleotemperature proxies for the eastern Indian Ocean, water column stratification over the NW Australian shelf (NSI) and low-latitude paleoproductivity (PP). **(a)** Nannofossil stratification index (NSI) at IODP Sites U1463 (in blue) and U1464 (orange) and TEX$_{86}$ derived temperature (°C) at IODP Site U1461 (He et al., 2021), located on the NW Australian shelf (grey). **(b)** NSI (as in panel **(a)**, but note reversed (r) scale of y-axis) and Mg/Ca-derived (*Trilobatus sacculifer*) SST (°C) at ODP Site 763 (Karas et al., 2011), located near the NW Australian shelf, but further offshore to the west on the Exmouth Plateau (grey stars). Horizontal black lines mark the suggested overall different paleoceanographic regimes before and after the Miocene-Pliocene boundary: Regime 1 with more stratified, weaker Leeuwin Current (LC) and Regime 2 with less stratified, stronger LC. **(c)** Standardised median PP in low latitudes in all major ocean basins (Karatsolis et al., 2022). Dashed lines indicate the 15.9 and 84.1 percentiles (i.e., the central 68.2 percentiles) of the compiled data. Black bars indicate (1) distinct decrease in mean and maximum coccolith length, as recorded in the Indian Ocean (Young, 1990) and (2) period of permanent shift to smaller sized nannofossils (Aubry 2007); dashed line illustrates (3) postulated LC intensity for the NW Australian shelf area in an earlier study (He et al., 2021). Vertical shadings indicate the stepwise decrease in NSI (brown), the decrease in low-latitude PP and nannofossil fluxes over the NW Australian shelf (green) and the small *Gephyrocapsa* dominance interval recorded in the Indian Ocean (grey; Young, 1990).

## 5. Conclusions

During the late Miocene to Pliocene, the NW Australian shelf area was influenced by an unrestricted ITF and dominated by warm tropical waters under humid climatic conditions. Nannofossil assemblages and paleotemperature gradients between the shelf region and the eastern Indian Ocean provide additional constraints on past regional ocean circulation and paleoproductivity between 6-3.5 Ma. The use of a nannofossil stratification index (NSI), which is based on the abundance of the most common calcareous nannofossil taxonomic groups, revealed a sustained shift in paleoceanographic regime across the Miocene-Pliocene boundary, between ~5.4-5.2 Ma. During that time, the water column over the NW Australian shelf transitioned from more stratified, tropical conditions (Regime 1; Figure 7) to a dynamic shelf environment with (seasonally) increased water column mixing and nutrient replenishment in the photic zone (Regime 2; Figure 7). This shift was possibly driven by more pronounced seasonal variations, expressed by intensified storm activity and eddy formation during the winter season and/or enhanced upwelling during the austral summer. The temperature gradients between the shelf area and the pelagic domains of the Indian Ocean, driven by a cooling over the shelf and a warming offshore, further strengthen this interpretation and suggest a more pronounced flow of the LC after ~5.2 Ma.  In addition to regional ocean circulation, we explored the changes in nannofossil assemblages that characterised the recently proposed termination of the late Miocene to early Pliocene biogenic bloom (4.6-4.4 Ma). Our data suggest that a restructuring of the nannofossil assemblage and an increase in diversity occurred during this interval, although the NSI was not significantly affected. The subsequent dominance of small *Gephyrocapsa* over the NW Australian shelf is interpreted as the expression of the broader dominance of this species in the Indian Ocean and its adaptation to the warm surface water conditions of the early Pliocene. This study highlights the differential response of the nannofossil assemblage to regional changes in the water column compared to broader-scale changes in paleoproductivity and hypothesized changes in the ocean's nutrient budget. However, the precise mechanisms that led to the proposed intensification in water column mixing and the possible link to an increase in seasonal extremes, remain unclear. Furthermore, more research is needed to shed light on the progressive dominance of small *Gephyrocapsa* during the early Pliocene, as well as the reasons for the changes in nannofossil assemblages that marked the low-latitude decrease in PP in other ocean basins.

**Author contributions**: B.T.K. conceived the study and performed the nannofossil counts and the data analysis, in close
consultation with J.H. B.T.K. interpreted the results and wrote the first draft of this manuscript. J.H. contributed to a critical interpretation and discussion of the results and helped with the writing of the manuscript, drafting of the figures and final revisions.

**Competing interests**: The authors declare that they have no conflict of interest.

**Acknowledgements**
This research used samples and data provided by the International Ocean Discovery Program and is part of the first author's PhD studies, financially supported by the Swedish Research Council (Vetenskapsrådet; Project Grant VR 2016-04434 to J. H.). We are grateful to the co-chiefs, shipboard science party, and crew of the *JOIDES Resolution* for the successful operations
during IODP Expedition 356.

**Data availability**
Additional data figures and tables that are part of this study can be found in the Appendix section. The raw data produced for this paper are publicly available at the Zenodo data repository (Karatsolis and Henderiks 2022,
https://doi.org/10.5281/zenodo.6965870).

**Appendix A: Supplementary data figures (nannofossil abundances and fluxes) and tables (paleotemperature gradients)**

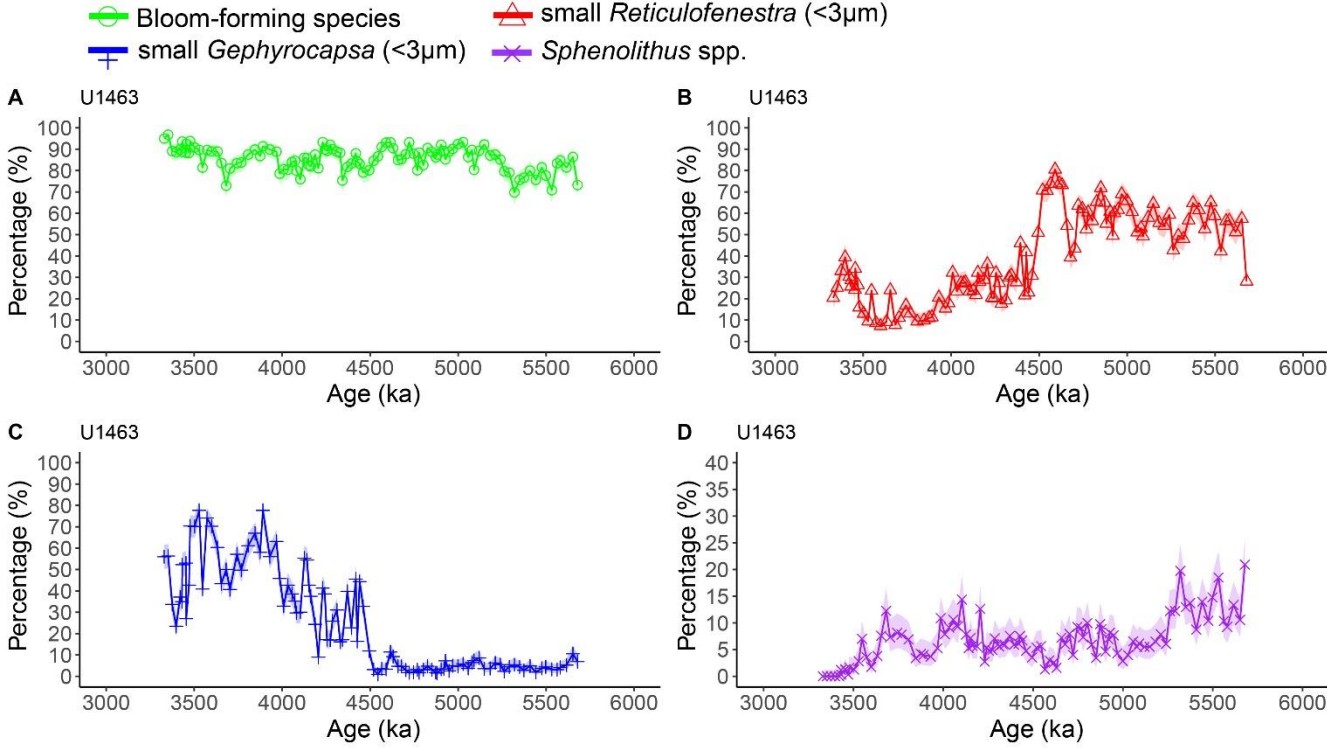

**Figure A1** Latest Miocene to early Pliocene relative abundances (%) of the dominant calcareous nannofossil species at IODP Site U1463. **(a)** Bloom-forming species (<5μm *Reticulofenestra* and <3μm *Gephyrocapsa* combined; green circles), **(b)** small (<3μm) *Reticulofenestra* (red triangles), **(c)** small (<3μm) *Gephyrocapsa* (blue plus signs), **(d)** *Sphenolithus* spp. (purple crosses). Shaded areas represent 95% confidence intervals (see methods). Note different y-axis scale in panel **(d)**.

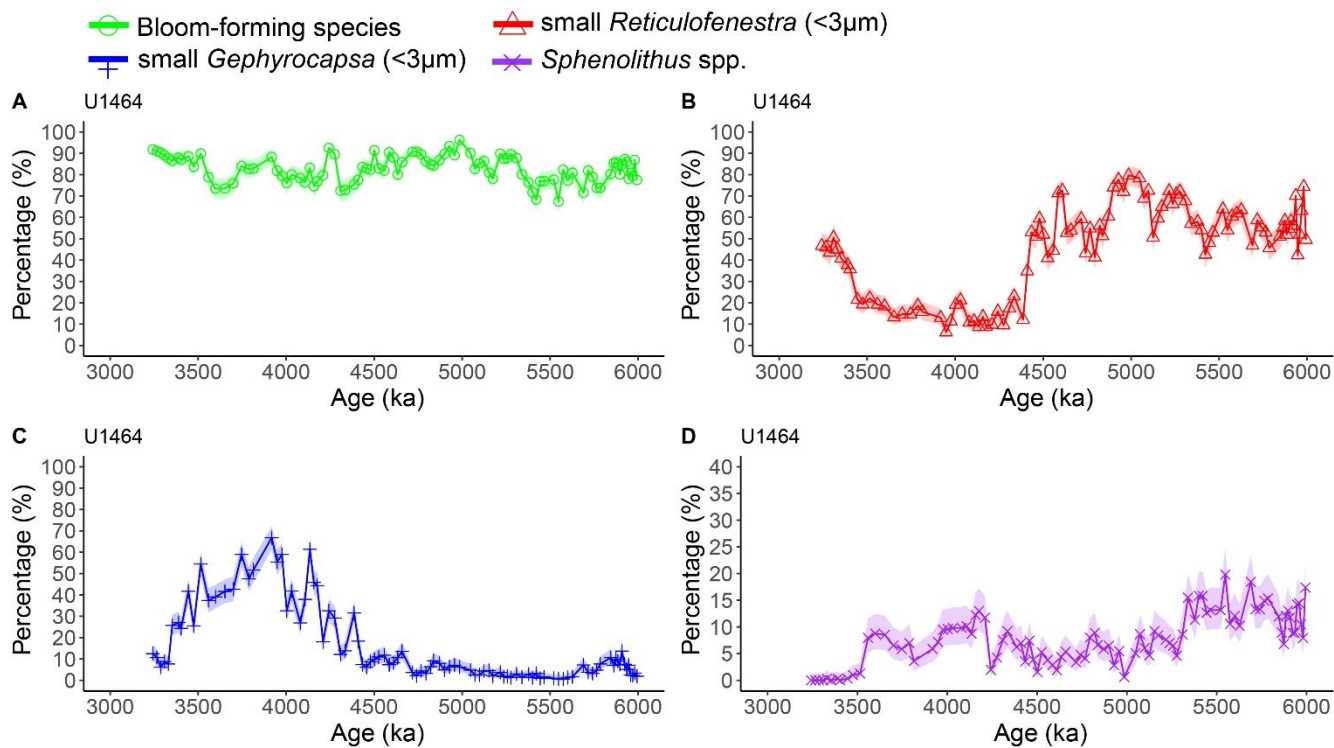

**Figure A2** Latest Miocene to early Pliocene relative abundances (%) of the dominant calcareous nannofossil species at IODP Site U1464. **(a)** Bloom-forming species (<5μm *Reticulofenestra* and <3μm *Gephyrocapsa* combined; green circles), **(b)** small (<3μm) *Reticulofenestra* (red triangles), **(c)** small (<3μm) *Gephyrocapsa* (blue plus signs), **(d)** *Sphenolithus* spp. (purple crosses). Shaded areas represent 95% confidence intervals (see methods). Note different y-axis scale in panel **(d)**.

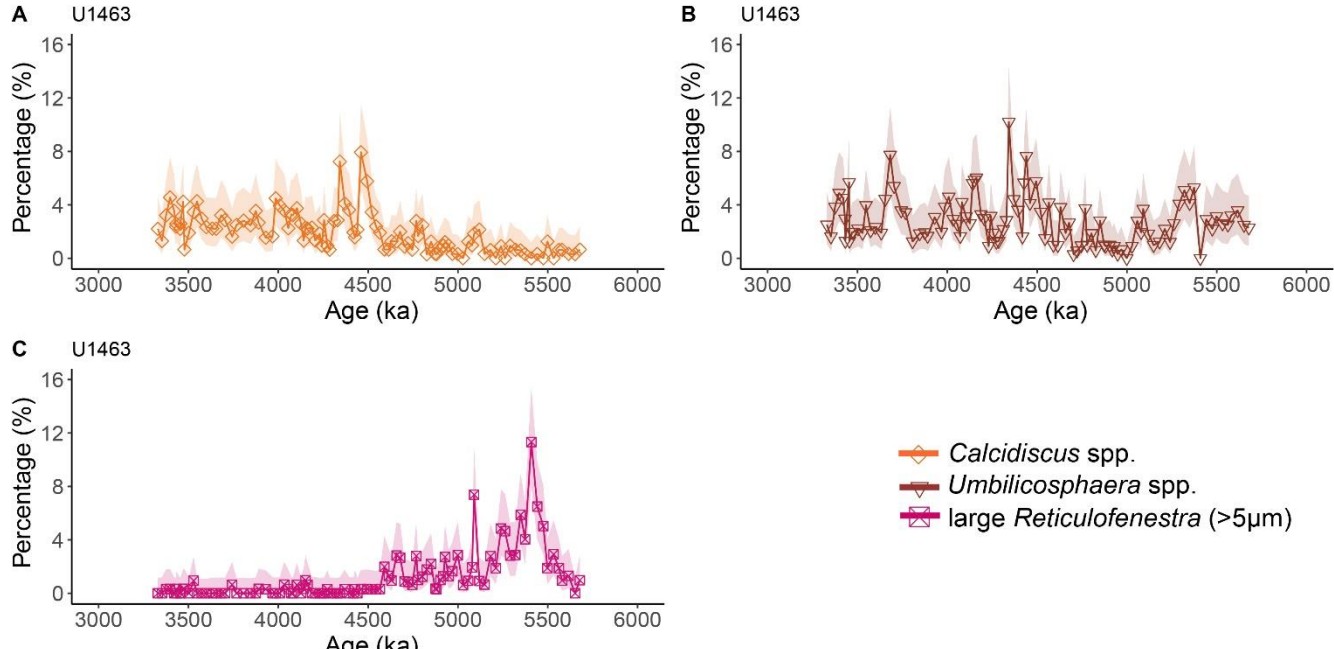

**Figure A3** Latest Miocene to early Pliocene relative abundances (%) of common calcareous nannofossil species at IODP Site U1463. **(a)** *Calcidiscus* spp. (orange diamonds), **(b)** *Umbilicosphaera* spp. (brown triangles) and **(c)** large (>5μm) *Reticulofenestra* (purple-pink squares). Shaded areas represent 95% confidence intervals (see methods).

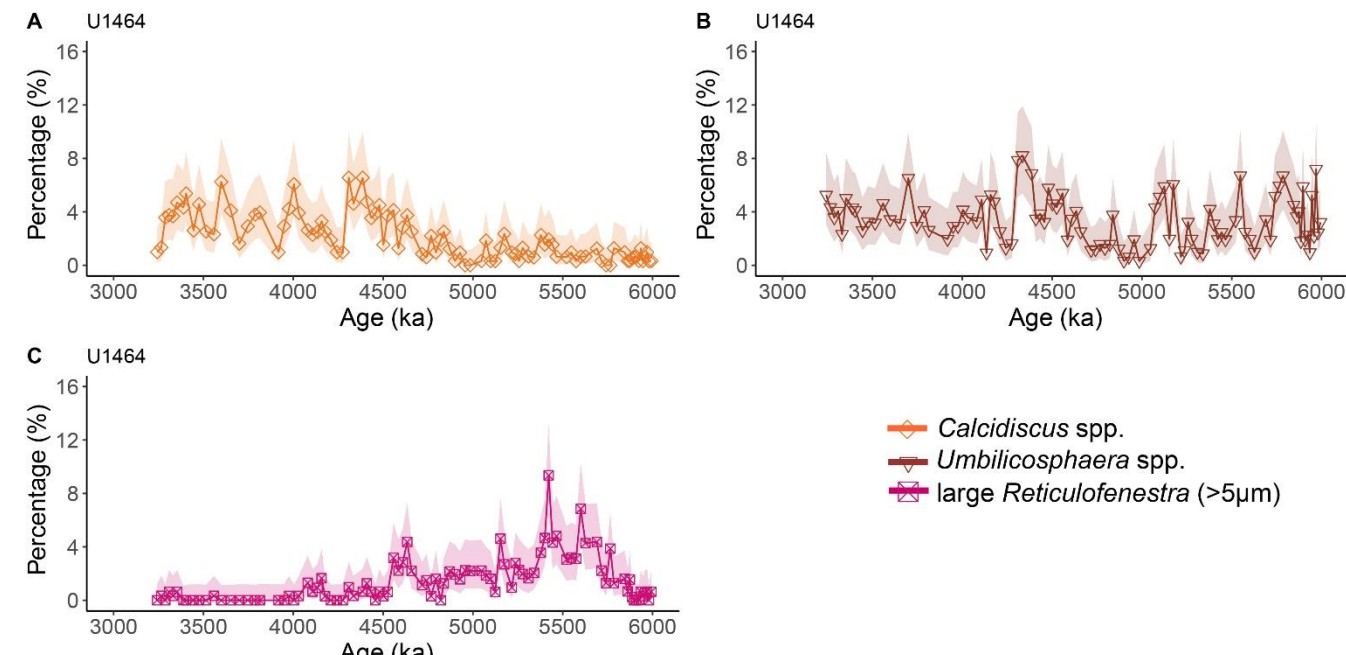

**Figure A4** Latest Miocene to early Pliocene relative abundances (%) of common calcareous nannofossil species at IODP Site U1464. **(a)** *Calcidiscus* spp. (orange diamonds), **(b)** *Umbilicosphaera* spp. (brown triangles) and **(c)** large (>5μm) *Reticulofenestra* (purple-pink squares). Shaded areas represent 95% confidence intervals (see methods).

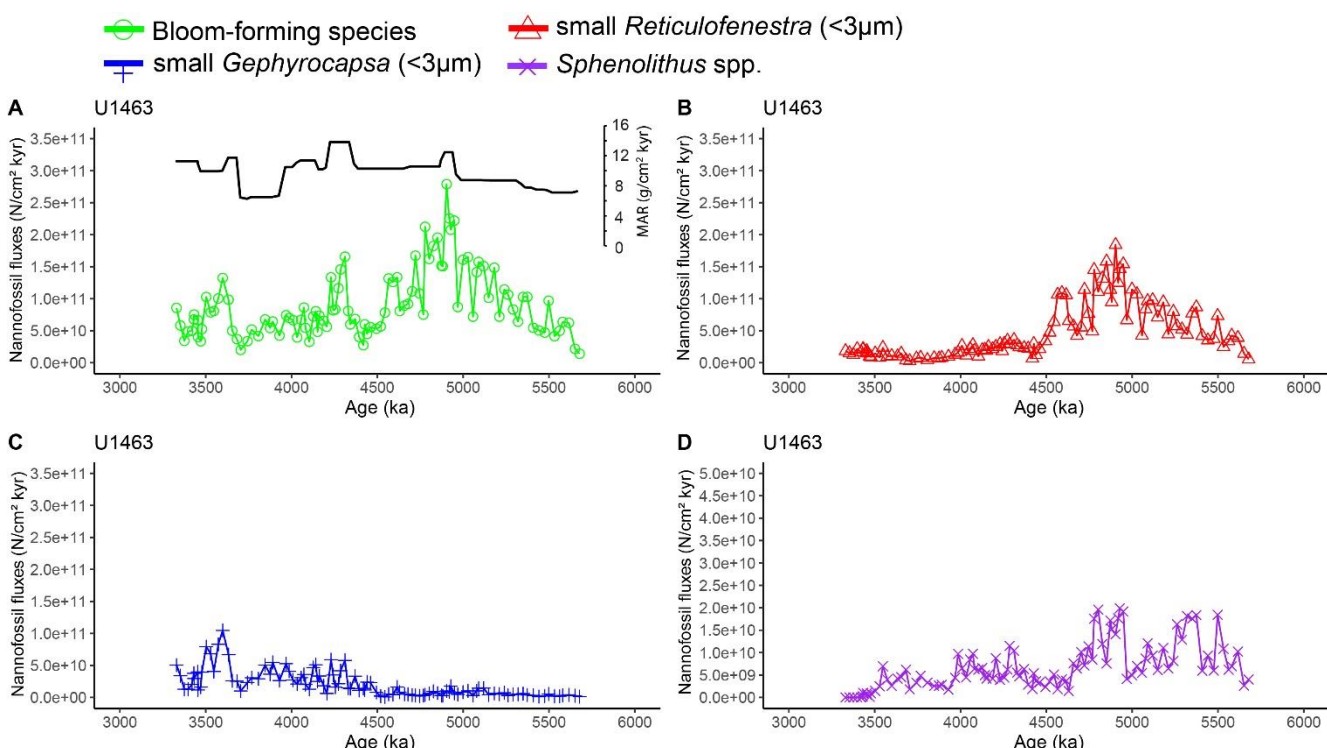

**Figure A5** Latest Miocene to early Pliocene nannofossil fluxes (NAR; N/cm$^2$ kyr) of commonly occurring calcareous nannofossil species at IODP Site U1463. **(a)** Bloom-forming species (<5μm *Reticulofenestra* and <3μm *Gephyrocapsa*; green circles), with inset bulk sediment MAR (g/cm$^2$ kyr) (black line); **(b)** small (<3μm) *Reticulofenestra* (red triangles), **(c)** small (<3μm) *Gephyrocapsa* (blue plus signs), **(d)** *Sphenolithus* spp. (purple crosses). Error bars of maximum replication error (15%; (Bordiga et al., 2015) were too small and therefore impaired the visual inspection of the figure. However, they are available as 595 part of the source data of the paper. Note different y-axis scale in panel **(d)**.

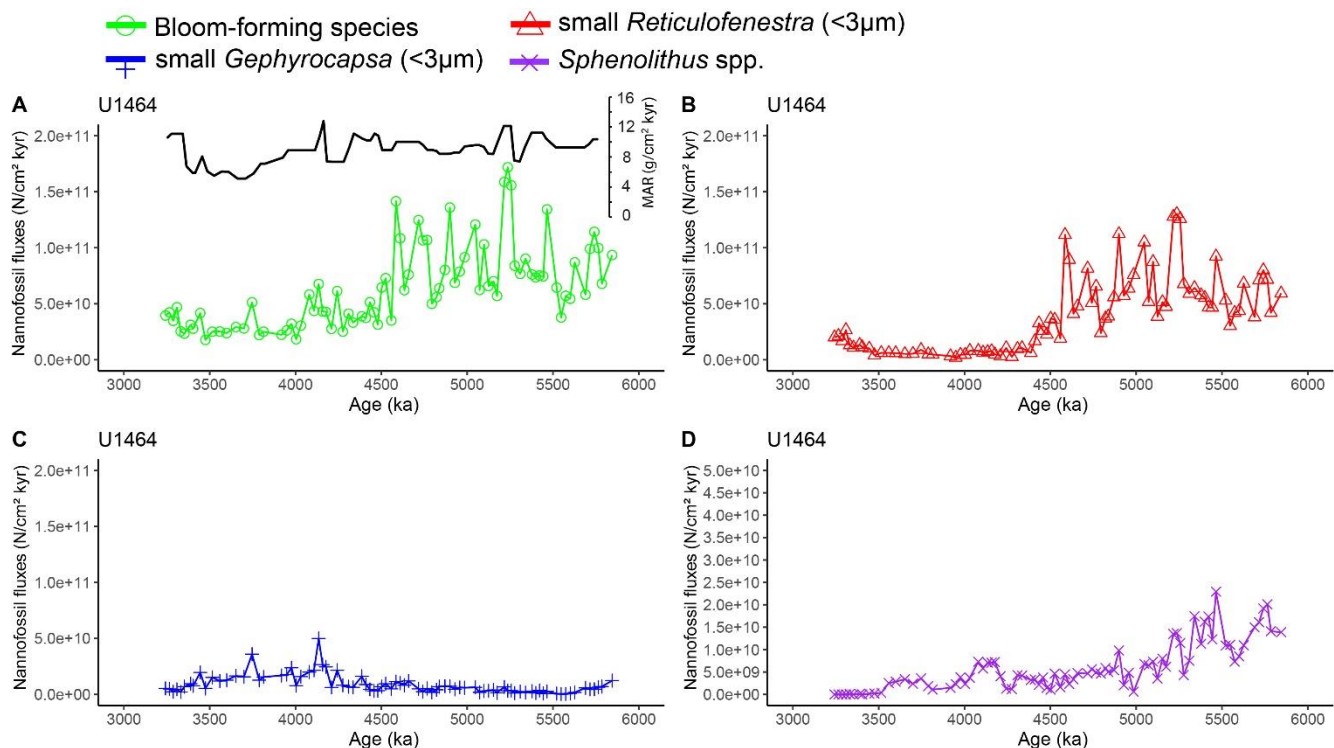

**Figure A6** Latest Miocene to early Pliocene nannofossil fluxes (NAR; N/cm$^2$ kyr) of commonly occurring calcareous nannofossil species at IODP Site U1464. **(a)** Bloom-forming species (<5µm *Reticulofenestra* and <3µm *Gephyrocapsa*; green circles), with inset bulk sediment MAR (g/cm$^2$ kyr) (black line), **(b)** small (<3µm) *Reticulofenestra* (red triangles), **(c)** small (<3µm) *Gephyrocapsa* (blue plus signs), **(d)** *Sphenolithus* spp. (purple crosses). Error bars of maximum replication error (15%; (Bordiga et al., 2015) were too small and therefore impaired the visual inspection of the figure. However, they are available as part of the source data of the paper. Note different y-axis scale in panel **(d)**.

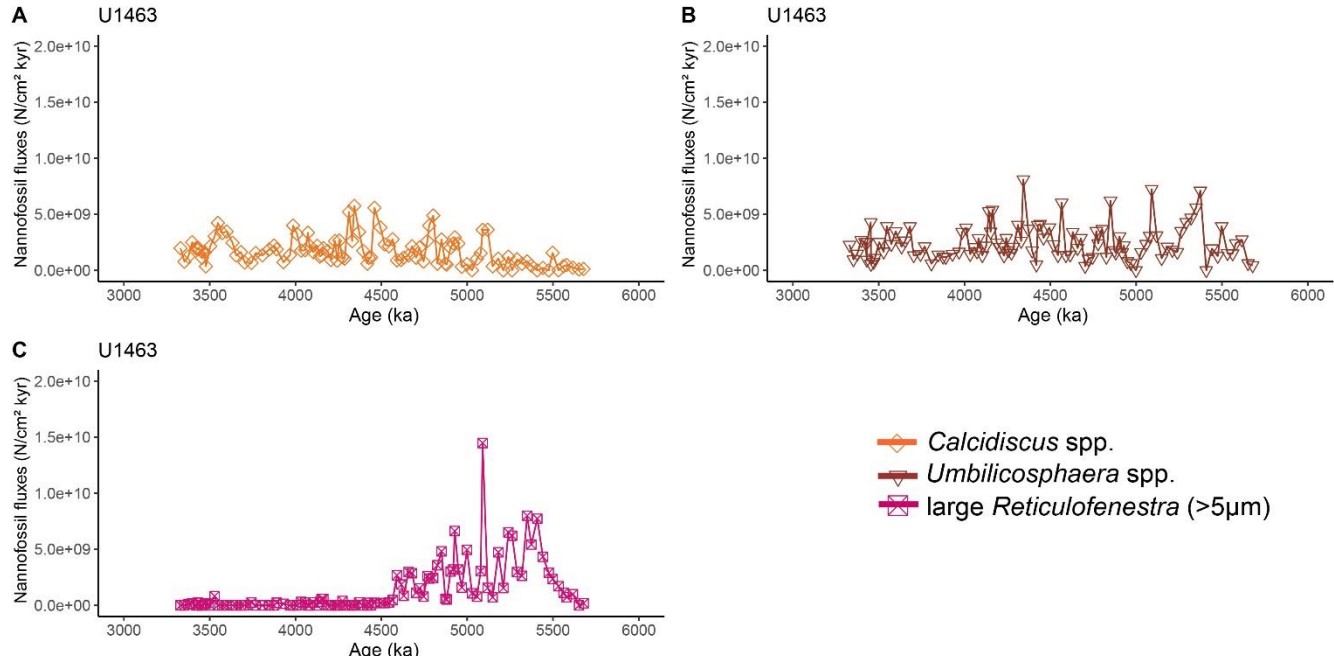

**Figure A7** Latest Miocene to early Pliocene nannofossil fluxes (NAR; N/cm² kyr) of commonly occurring calcareous nannofossil species at IODP Site U1463. **(a)** *Calcidiscus* spp. (orange diamonds), **(b)** *Umbilicosphaera* spp. (brown triangles) and **(c)** large (>5µm) *Reticulofenestra* (purple-pink squares). Error bars of maximum replication error (±15%; Bordiga et al., 610  2015) were too small and therefore impaired the visual inspection of the figure. However, they are available as part of the source data of this paper.

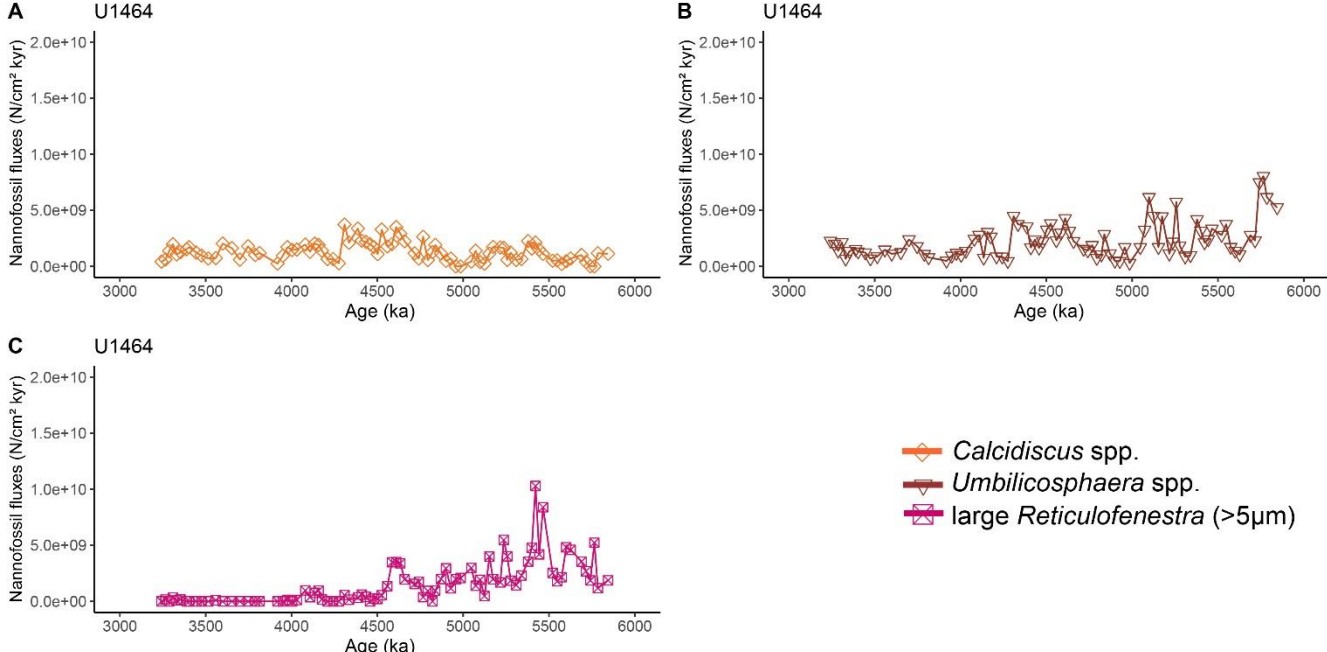

**Figure A8** Latest Miocene to early Pliocene nannofossil fluxes (NAR; N/cm$^2$ kyr) of commonly occurring calcareous nannofossil species at IODP Site U1464. **(a)** *Calcidiscus* spp. (orange diamonds), **(b)** *Umbilicosphaera* spp. (brown triangles) and **(c)** large (>5µm) *Reticulofenestra* (purple-pink squares). Error bars of maximum replication error (±15%; Bordiga et al., 2015) were too small and therefore impaired the visual inspection of the figure. However, they are available as part of the source data of this paper.

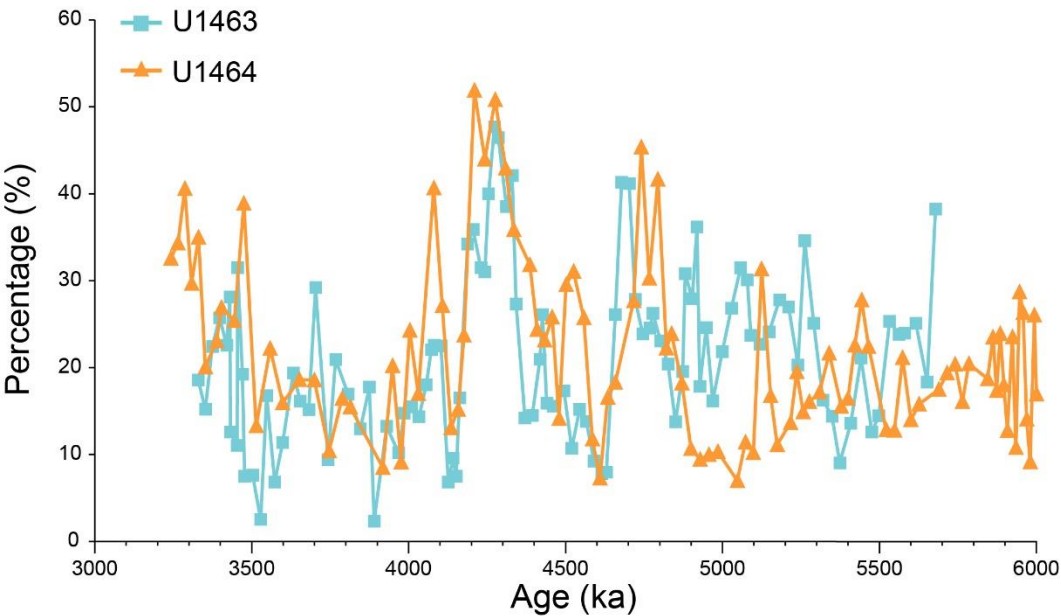

**Figure A9** Relative abundances (%) of medium-sized (3-5µm) *Reticulofenestra* species plotted against sample age (ka) at
IODP Sites U1463 (blue squares) and U1464 (orange triangles). Distinct peaks (exceeding 40% of the assemblage) can be
observed shortly before (~4.7 Ma) and after (~4.3 Ma) the 4.6–4.4 Ma decrease in PP.

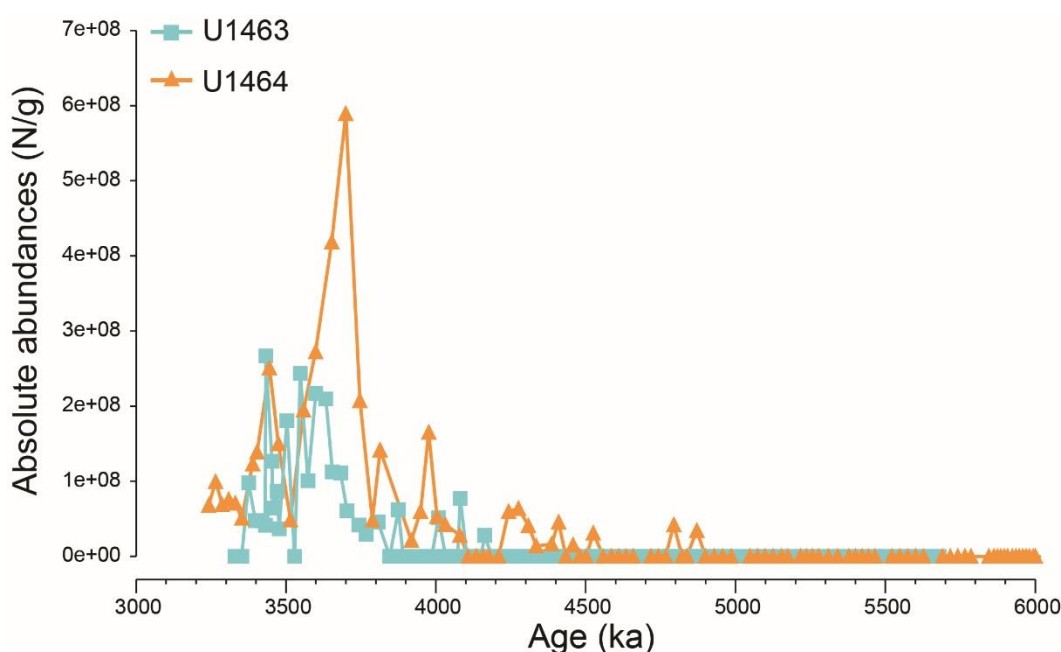

**Figure A10** Absolute coccolith abundances (N/g × 10$^8$) of *Pseudoemiliania lacunosa* plotted against sample age (ka) at IODP Sites U1463 (blue squares) and U1464 (orange triangles).

**Table A1** Summary of median differences in sample age (Δage, kyr) and paleotemperature (ΔT, °C) for the calculated gradients between eastern Indian Ocean and NW Australian shelf sites. Input data are alkenone- ($U^{k'}_{37}$) and GDGT-based (TEX$_{86}$) temperature estimates from IODP Site U1461 (He et al., 2021) and planktonic foraminifera-based (Mg/Ca) SST estimates for ODP Site 763 (Karas et al., 2011) and DSDP Site 214 (Karas et al., 2009).

| Gradient (Sites) | Median Δage* (kyr) 6-3.5 Ma | Median ΔT (°C) 5.2-3.5 Ma | Median ΔT (°C) 6-5.2 Ma |
|---|---|---|---|
| U1461 ( $U^{k'}_{37}$ )-763 (Mg/Ca) | 2.8 | -0.2 | |
| U1461 ( $U^{k'}_{37}$ )-214 (Mg/Ca) | 9.6 | -0.5 | |
| U1461 (TEX$_{86}$)-763 (Mg/Ca) | 2.9 | -0.6 | 2 |
| U1461 (TEX$_{86}$)-214 (Mg/Ca) | 7.8 | -1.15 | |

*Median Δage is calculated from the absolute difference between IODP Site U1461 sample ages and the closest available sample ages from ODP Site 763/DSDP Site 214

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
