# Peer review of "Late Neogene nannofossil assemblages as tracers of ocean circulation and paleoproductivity over the NW Australian shelf"

_Climate of the Past, 2022_

## Referee Comment (RC1)

In this manuscript, Karatsolis and Henderiks generated two calcareous nannofossil records from International Ocean Discovery Program (IODP) Sites U1463 and U1464, located in the NW Australian shelf, in order to reconstruct long-term changes in ocean circulation, seasonality and nutrient availability from ~ 6-3.5 million years ago (Ma). The authors characterised different periods of change in stratification and nutrient availability in the study area by analysing the shifts in the calcareous nannofossil dominant taxa and comparing them with palaeotemperature gradients between the NW Australian shelf and the eastern Indian Ocean.

Karatsolis and Henderiks found a marked regional change in the oceanographic conditions that affected the ecology of calcareous nannofossils across the Miocene to Pliocene boundary (5.4–5.2 Ma), which they attributed to an increase in seasonality and general intensification of the upper water column mixing. The authors also put the observed local variations in a more global context, considering events, such as the extinction of *Sphenolithus* spp. (~3.54 Ma) and the termination of the late Miocene to early Pliocene biogenic bloom in the eastern Indian Ocean (4.6-4.4 Ma).

**General comments**

This manuscript represents a substantial contribution to scientific progress within the scope of **Climate of the Past** and it is of interest for the coccolithophore, calcareous nannofossil, palaeoceanographic and micropalaeontological communities.

It is well written, logically structured, and presents a new calcareous nannofossil dataset.

The title reflects the contents of the manuscript and Karatsolis & Henderiks present an adequate summary of their work in the abstract. The state of the art and the main aims of this work are properly introduced in the first section.

The methods used in this piece of research seem adequate and are described section 2 of the manuscript. In my opinion, mathematical formulae, symbols, abbreviations and units are correctly defined and used through the text.

The interpretation and conclusions have been logically derived from their findings, and supported by the original data shown in section 3 (Results).

My main concern is that the authors should highlight more the variability between proxy types. They use different proxies, such as GDGT-based TEX86 temperature and alkenone-based $U37k'$ SST or Mg/Ca derived SSTs from *Trilobatus sacculifer*. I like that they use gradients, but in the manuscript, the uncertainty of working with different paleotemperature indicators need to be more clearly addressed. Perhaps adding some reference(s).

Figures and tables are in general clear (a very good example of this is Figure 7). I just have minor suggestions regarding the figures (see specific comments/technical corrections). I would recommend merging some of them (4 and 6).

I find that the references cited in the manuscript are adequate. I just found some typos.

The supplementary material is also adequate, but it could be improved (in some of the plots there are just too many wiggles). I would probably move some of the supplementary figures/plates to the main manuscript (see specific comments).

**Specific comments:**

**Abstract:**

Line (L.) 9-10: "and …and", replace by "as well as" to avoid repetition.

L.14: "and can therefore assist with more detailed reconstructions." sounds odd to me. Rephrase if possible.

L.14-25: In the abstract I would suggest to talk first about the general (more global) changes and the jump into the regional variations (or reword this part in a similar was as in the introduction). I understand the way that the authors try to "zoom out", but perhaps that makes more sense in the conclusions, rather than in the abstract. Therefore, my suggestion would be to reorganise this part.

L. 21: Could you provide more detail when you mention "Major changes"?

**1. Introduction:**

L. 28-34: It is confusing to the readers the way that the authors introduce the different terms "coccolithophores", "coccoliths", "nannoliths" and "nannofossils". Please rewrite this part.

L. 41: Please double check (here and elsewhere in the manuscript) that "equatorial warm water valve" is the right term. I have only encountered "equatorial valve" in the literature.

L. 48-49: Change to (/weaker)… (/El Niño)… (/3Sv)

L. 57-57: Sounds redundant. Double-check that sentence.

L. 61, 63: dot instead of circle

L. 63-64: Change to "The main surface oceanography of the Indo-Pacific region (dark red lines) and main path of the LC (lighter red line; adapted by (Auer et al., 2019; Gallagher et al., 2009) and the HC, which in this study are considered as one, are shown."

L. 65: The base map…

L. 71-76: Reword. Make it more concise.

L. 79: Mention somewhere what time Auer et al. (2019) previously covered.

L. 80: 100 km away

L. 92: Change to: (a) mixed layer and increased stratification (c) and SST…

L. 94: (b, d) there a space missing.

L. 103-105: As is Figure 2, I recommend putting (a), (b), etc before the description. E.g. (a) chlorophyll

**2. Material and Methods:**

L. 118: Specify seconds if possible.

L. 125, L. 135: Double-check the use of coccoliths /nannoliths etc here and elsewhere.

L. 129: Specify what is N (you do later in the text, but this is the first time).

L. 131: Make sure the comma is in the right place. It looks very close to the W, but I guess it is just a visual perception.

L. 139: Was the DBD calculated or downloaded from a specific site? Specify.

L. 139: considered for…specify

L. 141: were instead of " can still be"

L. 148-149: (column principal with ages representing distinct columns) is unclear to the reader.

L. 156: I would suggest using the whole name, Nannofossil Stratification Index.

L. 157: Write the whole name of the ratio and add (NSI) to introduce this term somewhere.

L. 160: Add a reference for that datum.

L. 182: I would delete or reword "Similarly, a ratio between *Reticulofenestra* species and *Florisphaera* has been used to monitor changes in the nutricline and thermocline during the Pleistocene (Flores et al., 2000)." This ratio is a bit different from what was previously mentioned…

L. 189: The authors should acknowledge the existing differences among different proxies to reconstruct the same environmental parameter (.e.g., SSTs) and reference it. Part of the last sentence (L. 200) should be moved up in this subsection.

This needs to be discussed further in the (sub-)section 3.4.

**3. Results:**

L. 205: How did the authors assessed the preservation? Expand or add a table in the supplementary material summarizing the ranking used. If it applies, reference it.

L. 228-229: Unclear to the reader. Rephrase, please.

L. 234-235: "NAR of *Sphenolithus* spp. bounced back to higher values, especially at IODP Site U1463" This is impossible to see in Figure A2. I would recommend the authors to space out the different taxa data in Figures A1 (especially in b and d) and A2 (all of them) using different Y-axes.

L. 241-242: Delete "As is the case for the nannofossil assemblage compositions", and change to: "Changes in NSI covary throughout the studied interval and correlate well between the two sites (Figure 4)."

L. 245: change demonstrates for shows?

L. 249. I suggest merging Figure 4 and 6.

L. 249-250: (light blue squares)… (orange triangles)

L. 255-257. I am not fully sure I understand this sentence. Are not assemblage and species composition the same? This sounds like circular thinking, but it is probably just a matter of rewording.

L. 278: Please use different symbols for the different sites.

L. 280: What does PP account for? I think it has not been introduced in the manuscript.

L. 285: "…Karas et al., 2011, upper…" (delete parenthesis)

**4. Discussion:**

L. 326-329: Is it possible for the authors to elaborate more in these higher and lower phases?

L. 345: "shelf area due to…" (I would mention it in the first sentence or merge L. 344-347).

L. 360: What about a combination of two or three of the 3 proposed mechanisms? Would that be an option? If so, perhaps add a sentence.

L. 371-372: "Shift" is mentioned twice in a sentence. Find a synonym.

L. 420-470. The reference Stuut et al. (2019) (https://doi.org/10.1029/2019GL083035) could be useful in the section 4.3 from the discussion because that study (in the continental margin of NW Australia) covers the last 5.3 Ma.

L. 423: Make sure PP is introduced before (not just in the figure captions), I guess the first time is in L. 67.

L. 478: *Trilobatus sacculifer*

L. 534-547: Figures A1 and A2. I already mentioned it, but I think it is difficult to see the data in some of the plots from these figure (especially if you print it in black and white). I would suggest using different Y axis.

When referring to a specific taxa, please add the symbol in the caption on top of the colour, as it was done for example in Figure C1. E.g., (<5µm *Reticulofenestra* and <3µm *Gephyrocapsa*; green circles).That will help colour-blind readers.

Also, I would probably include them as main figures (not as appendix), but this is up to the authors.

L. 537: Rather than error bars, I would use "shaded areas"

L. 545: 15%; Bordiga … (delete parenthesis).

In Figs. B1 and C1 the symbols of the legend do not match with the ones in the plot.

**References:**

There are several typos in the reference list. I spotted few, but the authors need to carefully check all the references.

L. 589: 2.45 Ma (space missing)

L. 594: (80-. )., (revise, something is missing here)

L. 595: K.-H.

L.598: $CO_2$ (subscript)

L. 606: 7(May) Double check

L. 614: Holloway

L. 626: 925, and… (delete comma?)

L.631: (August) Double check

L. 633: (80-. )., (revise, something is missing here)

L. 653: ,, (delete one comma)   (February) Double check

L. 657: Species names' in italics

L. 665-666: Add doi

L. 672: PAST

L. 675: $CO_2$ (subscript)

L. 684-686: Add doi

L. 693-694: Add doi

L. 695: Nye, H. .: (delete one dot)

L. 707-714: B-Th, B. −. T. or B. T.? The name of the main author is written in 3 different ways.

L. 756: Page numbers or number of pages?

L. 769: (80-. )., (revise, something is missing here)

---

## Author Comment (AC3)

Final Author Comment

Karatsolis et al. – cp-2022-60

We found the review comment by Mariem Saavedra (R1) very useful in several aspects. It helps us refine the presentation of the main points of the manuscript and clarify others, especially the ones regarding temperature proxy types. It also helps structure the manuscript in a clearer and more concise way. We therefore thank her warmly for her constructive review of our work.

*In this manuscript, Karatsolis and Henderiks generated two calcareous nannofossil records from International Ocean Discovery Program (IODP) Sites U1463 and U1464, located in the NW Australian shelf, in order to reconstruct long-term changes in ocean circulation, seasonality and nutrient availability from ~ 6-3.5 million years ago (Ma). The authors characterised different periods of change in stratification and nutrient availability in the study area by analysing the shifts in the calcareous nannofossil dominant taxa and comparing them with palaeotemperature gradients between the NW Australian shelf and the eastern Indian Ocean.*

*Karatsolis and Henderiks found a marked regional change in the oceanographic conditions that affected the ecology of calcareous nannofossils across the Miocene to Pliocene boundary (5.4–5.2 Ma), which they attributed to an increase in seasonality and general intensification of the upper water column mixing. The authors also put the observed local variations in a more global context, considering events, such as the extinction of Sphenolithus spp. (~3.54 Ma) and the termination of the late Miocene to early Pliocene biogenic bloom in the eastern Indian Ocean (4.6-4.4 Ma).*

*General comments*

*This manuscript represents a substantial contribution to scientific progress within the scope of Climate of the Past and it is of interest for the coccolithophore, calcareous nannofossil, palaeoceanographic and micropalaeontological communities.*

*It is well written, logically structured, and presents a new calcareous nannofossil dataset.*

*The title reflects the contents of the manuscript and Karatsolis & Henderiks present an adequate summary of their work in the abstract. The state of the art and the main aims of this work are properly introduced in the first section.*

*The methods used in this piece of research seem adequate and are described section 2 of the manuscript. In my opinion, mathematical formulae, symbols, abbreviations and units are correctly defined and used through the text.*

*The interpretation and conclusions have been logically derived from their findings, and supported by the original data shown in section 3 (Results).*

We thank the reviewer for her positive comments.

*My main concern is that the authors should highlight more the variability between proxy types. They use different proxies, such as GDGT-based TEX86 temperature and alkenone*

*based U37k´ SST or Mg/Ca derived SSTs from Trilobatus sacculifer. I like that they use gradients, but in the manuscript, the uncertainty of working with different paleotemperature indicators need to be more clearly addressed. Perhaps adding some reference(s).*

The uncertainty for working with different paleotemperature proxies will be clarified with additional information regarding the use of these proxies in Methodology section *2.4: Paleotemperature proxies and gradient calculation*, and the discussion section *4.2: Paleotemperature and inferred ocean circulation patterns*. We will include a set of references that highlight the previous use of these proxies in the area, but also globally, as well as the potential ambiguities that this may introduce when constructing gradients between them.

For example, we will further analyse the different interpretations of TEX86 as an SST versus whole water column indicator (De Vleeschouwer et al., 2019; Petrick et al., 2019; Smith et al., 2020; Smith et al., 2013) over shelfal areas of Australia.

*Figures and tables are in general clear (a very good example of this is Figure 7). I just have minor suggestions regarding the figures (see specific comments/technical corrections). I would recommend merging some of them (4 and 6).*

Detailed answers to figure suggestions are provided in the "specific comments" section.

*I find that the references cited in the manuscript are adequate. I just found some typos.*

Typos will be searched & corrected.

*The supplementary material is also adequate, but it could be improved (in some of the plots there are just too many wiggles). I would probably move some of the supplementary figures/plates to the main manuscript (see specific comments).*

Detailed answers to figure and supplementary figures suggestions are provided in the "specific comments" section.

*Specific comments:*

*Abstract:*

*Line (L.) 9-10: "and …and", replace by "as well as" to avoid repetition.*

We will follow this suggestion.

*L.14: "and can therefore assist with more detailed reconstructions." sounds odd to me. Rephrase if possible.*

This part of the sentence will be removed.

*L.14-25: In the abstract I would suggest to talk first about the general (more global) changes and the jump into the regional variations (or reword this part in a similar was as in the introduction). I understand the way that the authors try to "zoom out", but perhaps that makes more sense in the conclusions, rather than in the abstract. Therefore, my suggestion would be to reorganise this part.*

We will attend to stream-lining the abstract taking the reviewers' comments into consideration.

*L. 21: Could you provide more detail when you mention "Major changes"?*

More detail will be provided by replacing "Major changes" with "Significant changes in nannofossil species distribution and abundances". Examples of what these significant changes consisted of is also provided later in this sentence "*such as the extinction of Sphenolithus spp. (~3.54 Ma) and the termination of the late Miocene to early Pliocene biogenic bloom in the eastern Indian Ocean (4.6-4.4 Ma), occurred long after this regional regime shift.*"

*1. Introduction:*

*L. 28-34: It is confusing to the readers the way that the authors introduce the different terms "coccolithophores", "coccoliths", "nannoliths" and "nannofossils". Please rewrite this part.*

We will clarify (and simplify) this introduction. We will keep (calcareous) nannofossils as the main term to be used throughout the manuscript, since it includes both the coccoliths and the nannoliths.

*L. 41: Please double check (here and elsewhere in the manuscript) that "equatorial warm water valve" is the right term. I have only encountered "equatorial valve" in the literature.*

This is a nonstandard term, so we expect that it has not been used in the literature. However, since we would like to pinpoint that this is the only equatorial valve that allows the flow of major warm water currents nowadays, we will keep this term and add quotation marks.

*L. 48-49: Change to (/weaker)… (/El Niño)… (/3Sv)*

We will follow this suggestion.

*L. 57-57: Sounds redundant. Double-check that sentence.*

The redundant part of the sentence will be removed.

*L. 61, 63: dot instead of circle*

"Circle" will be replaced with "dot"

*L. 63-64: Change to "The main surface oceanography of the Indo-Pacific region (dark red lines) and main path of the LC (lighter red line; adapted by (Auer et al., 2019; Gallagher et al., 2009) and the HC, which in this study are considered as one, are shown."*

We will follow this suggestion.

*L. 65: The base map…*

"The" will be added.

*L. 71-76: Reword. Make it more concise.*

We will follow this suggestion.

*L. 79: Mention somewhere what time Auer et al. (2019) previously covered.*

We will add this within the reference brackets.

*L. 80: 100 km away*

Noted.

*L. 92: Change to: (a) mixed layer and increased stratification (c) and SST…*

We will follow this suggestion.

*L. 94: (b, d) there a space missing.*

A space will be added.

*L. 103-105: As is Figure 2, I recommend putting (a), (b), etc before the description. E.g. (a) chlorophyll*

We will follow this suggestion.

2. Material and Methods:

*L. 118: Specify seconds if possible.*

The seconds will be added (30-60 seconds).

*L. 125, L. 135: Double-check the use of coccoliths /nannoliths etc here and elsewhere.*

We will double-check the use of these terms as suggested and follow terms as introduced in revised Introduction.

*L. 129: Specify what is N (you do later in the text, but this is the first time).*

We will specify what N stands for when using this abbreviation for the first time.

*L. 131: Make sure the comma is in the right place. It looks very close to the W, but I guess it is just a visual perception.*

We will check if the comma following the equation is in the right place.

*L. 139: Was the DBD calculated or downloaded from a specific site? Specify.*

DBD was calculated for the respective holes from the information provided in the LIMS database of IODP. Since values were not changing significantly across cores, an average of the available data for the interval of interest was calculated, and then used to calculate nannofossil accumulation rates (NAR). We will specify this information in the text.

*L. 139: considered for…specify*

We will specify this to read "Nannofossil flux records older than 5.8 Ma at IODP Site U1464 were not included in the interpretation of changes in nannofossil abundance".

*L. 141: were instead of " can still be"*

"Can still be" will be replaced with "were".

*L. 148-149: (column principal with ages representing distinct columns) is unclear to the reader.*

Indeed, this is a confusing technical detail that does not add to the description of the method. We will remove it.

*L. 156: I would suggest using the whole name, Nannofossil Stratification Index.*

We will use the full name of the index.

*L. 157: Write the whole name of the ratio and add (NSI) to introduce this term somewhere.*

We will follow this suggestion.

*L. 160: Add a reference for that datum.*

The reference of the datum will be added.

*L. 182: I would delete or reword "Similarly, a ratio between Reticulofenestra species and Florisphaera has been used to monitor changes in the nutricline and thermocline during the Pleistocene (Flores et al., 2000)." This ratio is a bit different from what was previously mentioned…*

Good point. These two ratios were not used to infer the same conditions. We will remove this analogy.

*L. 189: The authors should acknowledge the existing differences among different proxies to reconstruct the same environmental parameter (.e.g., SSTs) and reference it. Part of the last sentence (L. 200) should be moved up in this subsection.*

This is a very valid point. The three different paleotemperature proxies have been previously used to infer paleotemperatures from various depths of the water column, from a variety of different environments and with various seasonal and geographical biases. As we mentioned in our response to the "general comments", we will present in more detail the differences among TEX86, U37k' and Mg/Ca reconstructions. We will mainly focus on the previous use of these proxies in shelfal areas of Australia and the tropical Indian Ocean during the Miocene and Pliocene, as presented by the studies where the paleotemperature information were retrieved. The relevant references will be added.

For the same purpose, we will follow the suggestion of moving L.200 up in this subsection.

*This needs to be discussed further in the (sub-)section 3.4.*

Having acknowledged the differences between proxies, we will further discuss the use of the calculated gradients, the information they can provide and their limitations in section 3.4: *Paleotemperature gradients*. We will also integrate this discussion to the reasoning behind the different paleotemperature gradient labelling (ΔT versus ΔSST).

*3. Results:*

*L. 205: How did the authors assessed the preservation? Expand or add a table in the supplementary material summarizing the ranking used. If it applies, reference it.*

Preservation in samples from U1463 and U1464 was assessed in the initial reports of Expedition 356 and followed a 1-5 scoring system (1 = poor [P], 2 = moderate, 3 = good [G], 4 = very good), following the definitions from Gallagher et al., (2017a, 2017b; Expedition 356 site summaries). In the initial reports, nannofossil preservation was assessed to score 2 and 3, demonstrating moderate and in cases

good preservation. In this study, through visual inspection, we observed that good preservation was actually rare and therefore labelled the preservation for the studied interval as moderate (score 2). We will expand on the matter in L.205 and reference the score system used.

*L. 228-229: Unclear to the reader. Rephrase, please.*

We will rephrase this sentence and make it clearer.

*L. 234-235: "NAR of Sphenolithus spp. bounced back to higher values, especially at IODP Site U1463" This is impossible to see in Figure A2. I would recommend the authors to space out the different taxa data in Figures A1 (especially in b and d) and A2 (all of them) using different Y-axes.*

We will follow this suggestion and use different axes in Figures A1 and A2, in order to make the species relative abundance and NAR values more visible.

*L. 241-242: Delete "As is the case for the nannofossil assemblage compositions", and change to: "Changes in NSI covary throughout the studied interval and correlate well between the two sites (Figure 4)."*

We will follow this suggestion and substitute the sentence.

*L. 245: change demonstrates for shows?*

We will substitute this word.

*L. 249. I suggest merging Figure 4 and 6.*

We will follow this suggestion and merge Figure 4 and 6. To better accommodate this change, sub-sections 2.2 and 2.3 will be merged under the title: 3.2 *Nannofossil multivariate analysis*.

*L. 249-250: (light blue squares)… (orange triangles)*

We will add this information.

*L. 255-257. I am not fully sure I understand this sentence. Are not assemblage and species composition the same? This sounds like circular thinking, but it is probably just a matter of rewording.*

We will rephrase this sentence to read "*By color-coding the samples based on their age across the main time intervals of interest (5.4-5.2 Ma and after 4.6–4.4 Ma; Figure 5), we test if any of the observed changes in the abundance of dominant species were accompanied by larger shifts in the relationship between relative abundances of species in multivariate space.*"

*L. 278: Please use different symbols for the different sites.*

We will replot this figure, using different symbols for the different sites.

*L. 280: What does PP account for? I think it has not been introduced in the manuscript.*

PP stands for paleoproductivity. We will introduce this term in the manuscript before using the abbreviation.

*L. 285: "...Karas et al., 2011, upper…" (delete parenthesis)*

The parenthesis will be deleted.

*4. Discussion:*

*L. 326-329: Is it possible for the authors to elaborate more in these higher and lower phases?*

We have described these phases in detail in section 3.2: *Nannofossil stratification index (NSI)*

*"After ~4.2 Ma, this surface water stratification proxy shows higher variation with short intervals of higher values (4.2-4 Ma and 3.8-3.54 Ma), that correspond to peaks in relative abundance of Sphenolithus spp. Between 4 and 3.8 Ma, NSI demonstrates low values…"*

This sentence will be written more concisely to read: "*"After ~4.2 Ma, NSI shows higher variation with short intervals of higher values (4.2-4 Ma and 3.8-3.54 Ma). These intervals correspond to peaks in relative abundance of Sphenolithus spp.*"

Additionally, in the discussion, we will elaborate more on the possible reasons that might have led to higher variability between higher and lower phases of NSI during this interval. Although the mechanism is unclear, it is interesting that higher amplitude NSI after 4.2 Ma, correlate well with higher amplitude changes observed in the temperature gradients after ~4.3 Ma. This would suggest a similar controlling mechanism to the one we are proposing in this manuscript, but in this case on shorter timescales.

*L. 345: "shelf area due to…" (I would mention it in the first sentence or merge L. 344-347).*

We will follow this suggestion and merge L. 344-347.

*L. 360: What about a combination of two or three of the 3 proposed mechanisms? Would that be an option? If so, perhaps add a sentence.*

We will add a sentence mentioning that any possible combination of the proposed mechanisms could have had similar results.

*L. 371-372: "Shift" is mentioned twice in a sentence. Find a synonym.*

The first "shift" will be replaced with "change".

*L. 420-470. The reference Stuut et al. (2019) (https://doi.org/10.1029/2019GL083035) could be useful in the section 4.3 from the discussion because that study (in the continental margin of NW Australia) covers the last 5.3 Ma.*

We will add this reference in the beginning of the paragraph, as part of the overall paleoclimatic regime in the continental margin of NW Australia during the early Pliocene before we focus on paleoproductivity and paleoecology matters.

*L. 423: Make sure PP is introduced before (not just in the figure captions), I guess the first time is in L. 67.*

We will make sure that this abbreviation is introduced in the text.

*L. 478: Trilobatus sacculifer*

We will write the full name of the genus.

*L. 534-547: Figures A1 and A2. I already mentioned it, but I think it is difficult to see the data in some of the plots from these figure (especially if you print it in black and white). I would suggest using different Y axis.*

As mentioned earlier, we will follow the suggestion of giving the plots some space, by modifying the y-axis.

*When referring to a specific taxa, please add the symbol in the caption on top of the colour, as it was done for example in Figure C1. E.g., (<5µm Reticulofenestra and <3µm Gephyrocapsa; green circles).That will help colour-blind readers.*

We will follow this suggestion.

*Also, I would probably include them as main figures (not as appendix), but this is up to the authors.*

We will consider this suggestion, although we think that the figures contain a lot of information (as you probably already noticed), which are not all presented in detail in the main manuscript and would be tangential to the main focus of this manuscript. We would like to keep a "grouping-oriented" strategy by focusing mainly on dominant species and ratios between them. Yet, we want to show all (raw) data that is included in the CA and Shannon-index calculations for transparency and for those seeking more details, in the supplement.

*L. 537: Rather than error bars, I would use "shaded areas"*

Good point. We will use the term "shaded areas".

*L. 545: 15%; Bordiga … (delete parenthesis).*

We will delete this parenthesis.

*In Figs. B1 and C1 the symbols of the legend do not match with the ones in the plot.*

Good observation, thanks. We will correct the symbols so that they match the plot.

*References:*

*There are several typos in the reference list. I spotted few, but the authors need to carefully check all the references.*

*L. 589: 2.45 Ma (space missing)*

*L. 594: (80-. )., (revise, something is missing here)*

*L. 595: K.-H.*

*L.598: $CO_2$ (subscript)*

*L. 606: 7(May) Double check*

*L. 614: Holloway*

*L. 626: 925, and… (delete comma?)*

*L.631: (August) Double check*

*L. 633: (80-. )., (revise, something is missing here)*

*L. 653: ,, (delete one comma) (February) Double check*

*L. 657: Species names' in italics*

*L. 665-666: Add doi*

*L. 672: PAST*

*L. 675: CO$_2$ (subscript)*

*L. 684-686: Add doi*

*L. 693-694: Add doi*

*L. 695: Nye, H. .: (delete one dot)*

*L. 707-714: B-Th, B. –. T. or B. T.? The name of the main author is written in 3 different ways.*

*L. 756: Page numbers or number of pages?*

*L. 769: (80-. )., (revise, something is missing here)*

We thank the reviewer for taking the time to identify these typos. The references will be revised accordingly.

References

De Vleeschouwer, D., et al., 2019. Stepwise weakening of the Pliocene Leeuwin Current. Geophys. Res. Lett. 46, 8310–8319.

Gallagher, S.J., et al., 2017. Site U1463. In Gallagher, S.J., Fulthorpe, C.S., Bogus, K., and the Expedition 356 Scientists, Indonesian Throughflow. Proceedings of the International Ocean Discovery Program, 356: College Station, TX (International Ocean Discovery Program).

Gallagher, S.J., et al., 2017. Site U1464. In Gallagher, S.J., Fulthorpe, C.S., Bogus, K., and the Expedition 356 Scientists, Indonesian Throughflow. Proceedings of the International Ocean Discovery Program, 356: College Station, TX (International Ocean Discovery Program).

Petrick, B., et al., 2019. Glacial Indonesian Throughflow weakening across the Mid-Pleistocene Climatic Transition. Sci. Rep. 9, 16995.

Smith, M., et al., 2013. Comparison of $U_{37}^{k'}$, TEX$_{86}$ and LDI temperature proxies for reconstruction of South-East Australian Ocean temperatures. Org. Geochem. 64, 94–104.

Smith, R.A., et al., 2020. Plio-Pleistocene Indonesian Throughflow variability drove Eastern Indian Ocean sea surface temperatures. Paleoceanogr. Paleoclimatol. 35, e2020PA003872.

---

## Author Response (AR1)

**Author's response (R1)**

Karatsolis et al. – cp-2022-60

We found the review comment by Mariem Saavedra (R1) very useful in several aspects. It helps us refine the presentation of the main points of the manuscript and clarify others, especially the ones regarding temperature proxy types. It also helps structure the manuscript in a clearer and more concise way. We therefore thank her warmly for her constructive review of our work.

*Line numbers refer to lines in the track changes document

*In this manuscript, Karatsolis and Henderiks generated two calcareous nannofossil records from International Ocean Discovery Program (IODP) Sites U1463 and U1464, located in the NW Australian shelf, in order to reconstruct long-term changes in ocean circulation, seasonality and nutrient availability from ~ 6-3.5 million years ago (Ma). The authors characterised different periods of change in stratification and nutrient availability in the study area by analysing the shifts in the calcareous nannofossil dominant taxa and comparing them with palaeotemperature gradients between the NW Australian shelf and the eastern Indian Ocean.*

*Karatsolis and Henderiks found a marked regional change in the oceanographic conditions that affected the ecology of calcareous nannofossils across the Miocene to Pliocene boundary (5.4–5.2 Ma), which they attributed to an increase in seasonality and general intensification of the upper water column mixing. The authors also put the observed local variations in a more global context, considering events, such as the extinction of Sphenolithus spp. (~3.54 Ma) and the termination of the late Miocene to early Pliocene biogenic bloom in the eastern Indian Ocean (4.6-4.4 Ma).*

*General comments*

*This manuscript represents a substantial contribution to scientific progress within the scope of Climate of the Past and it is of interest for the coccolithophore, calcareous nannofossil, palaeoceanographic and micropalaeontological communities.*

*It is well written, logically structured, and presents a new calcareous nannofossil dataset.*

*The title reflects the contents of the manuscript and Karatsolis & Henderiks present an adequate summary of their work in the abstract. The state of the art and the main aims of this work are properly introduced in the first section.*

*The methods used in this piece of research seem adequate and are described section 2 of the manuscript. In my opinion, mathematical formulae, symbols, abbreviations and units are correctly defined and used through the text.*

*The interpretation and conclusions have been logically derived from their findings,and supported by the original data shown in section 3 (Results).*

We thank the reviewer for her positive comments.

*My main concern is that the authors should highlight more the variability between proxy types. They use different proxies, such as GDGT-based TEX86 temperature and alkenone based U37k´ SST or Mg/Ca derived SSTs from Trilobatus sacculifer. I like that they use gradients, but in the manuscript, the uncertainty of working with different paleotemperature indicators need to be more clearly addressed. Perhaps adding some reference(s).*

The uncertainty of working with different paleotemperature proxies is clarified with additional information regarding the temperature estimates that each proxy provides, as well as relevant references of their previous use, mainly in Methodology section *2.4: Paleotemperature proxies and gradient calculation* (lines 211-241).

*Figures and tables are in general clear (a very good example of this is Figure 7). I just have minor suggestions regarding the figures (see specific comments/technical corrections). I would recommend merging some of them (4 and 6).*

Detailed answers to figure suggestions are provided in the "specific comments" section.

*I find that the references cited in the manuscript are adequate. I just found some typos.*

Typos were searched & corrected.

*The supplementary material is also adequate, but it could be improved (in some of the plots there are just too many wiggles). I would probably move some of the supplementary figures/plates to the main manuscript (see specific comments).*

Detailed answers to figure and supplementary figures suggestions are provided in the "specific comments" section.

*Specific comments:*

*Abstract:*

*Line (L.) 9-10: "and …and", replace by "as well as" to avoid repetition.*

We replaced the second "and" with "including" (line 13).

*L.14: "and can therefore assist with more detailed reconstructions." sounds odd to me. Rephrase if possible.*

This part of the sentence has been removed.

*L.14-25: In the abstract I would suggest to talk first about the general (more global) changes and the jump into the regional variations (or reword this part in a similar was as in the introduction). I understand the way that the authors try to "zoom out", but perhaps that makes more sense in the conclusions, rather than in the abstract. Therefore, my suggestion would be to reorganise this part.*

There is barely mention of general (more global) changes when establishing the setting of the study in the current abstract, and as such it goes straight to the (more regional) point. However, we have now streamlined some of its parts by taking away a repetitive part (re. sedimentary archives in first sentence and the IODP declarations) and shortening in parts for a better flow (lines 7-30).

*L. 21: Could you provide more detail when you mention "Major changes"?*

More detail was provided by replacing "Major changes" with "Significant changes in nannofossil abundance and species composition". Examples of what these significant changes consisted of is also provided later in this sentence "*such as the termination of the late Miocene to early Pliocene biogenic bloom in the eastern Indian Ocean (4.6-4.4 Ma) and the extinction of Sphenolithus spp. (~3.54 Ma), occurred long after this regional regime shift*." (lines 26-27)

*1. Introduction:*

*L. 28-34: It is confusing to the readers the way that the authors introduce the different terms "coccolithophores", "coccoliths", "nannoliths" and "nannofossils". Please rewrite this part.*

We have clarified (and simplified) this introduction, keeping (calcareous) and nannofossils (nannoplankton) as the main term used throughout the manuscript, since it includes both the coccoliths and the nannoliths (lines 34-50).

*L. 41: Please double check (here and elsewhere in the manuscript) that "equatorial warm water valve" is the right term. I have only encountered "equatorial valve" in the literature.*

This is a nonstandard term, so we expect that it has not been used in the literature. However, since we would like to pinpoint that this is the only equatorial valve that allows the flow of major warm water currents nowadays, we kept this term and added quotation marks.

*L. 48-49: Change to (/weaker)… (/El Niño)… (/3Sv)*

We have followed this suggestion (lines 59-60).

*L. 57-57: Sounds redundant. Double-check that sentence.*

The redundant part of the sentence has been removed.

*L. 61, 63: dot instead of circle*

"Circle" has been replaced with "dot" (line 73).

*L. 63-64: Change to "The main surface oceanography of the Indo-Pacific region (dark red lines) and main path of the LC (lighter red line; adapted by (Auer et al., 2019; Gallagher et al., 2009) and the HC, which in this study are considered as one, are shown."*

We have followed this suggestion (lines 76-78).

*L. 65: The base map…*

"The" has been added.

*L. 71-76: Reword. Make it more concise.*

We have followed this suggestion (lines 86-89).

*L. 79: Mention somewhere what time Auer et al. (2019) previously covered.*

We have added the time interval studied by Auer et al. (2019) within the reference brackets.

*L. 80: 100 km away*

Noted.

*L. 92: Change to: (a) mixed layer and increased stratification (c) and SST...*

The figure caption has been restructured taking this comment into consideration (108-109).

*L. 94: (b, d) there a space missing.*

The caption has been restructured and a space is not needed anymore.

*L. 103-105: As is Figure 2, I recommend putting (a), (b), etc before the description. E.g. (a) chlorophyll*

This Figure caption has been restructured accordingly (lines 119-120).

2. Material and Methods:

*L. 118: Specify seconds if possible.*

The seconds have been added (30-60 seconds; line 137).

*L. 125, L. 135: Double-check the use of coccoliths /nannoliths etc here and elsewhere.*

We double-checked the use of these terms as suggested and followed terms as introduced in the revised Introduction.

*L. 129: Specify what is N (you do later in the text, but this is the first time).*

We have specified what N stands for (line 148).

*L. 131: Make sure the comma is in the right place. It looks very close to the W, but I guess it is just a visual perception.*

The comma following the equation seems to be in the right place.

*L. 139: Was the DBD calculated or downloaded from a specific site? Specify.*

DBD was calculated for the respective holes from the information provided in the LIMS database of IODP. Since values were not changing significantly across cores, an average of the available data for the interval of interest was calculated, and then used to calculate nannofossil accumulation rates (NAR). We have now specified this information in the text (lines 157-159).

*L. 139: considered for...specify*

We have now specified this to read "Nannofossil flux records older than 5.8 Ma at IODP Site U1464 were not included in the interpretation of changes in nannofossil abundance".

*L. 141: were instead of " can still be"*

"Can still be" was replaced with "were".

*L. 148-149: (column principal with ages representing distinct columns) is unclear to the reader.*

Indeed, this is a confusing technical detail that does not add to the description of the method. We have now removed it.

*L. 156: I would suggest using the whole name, Nannofossil Stratification Index.*

The full name of the index is now used (line 179).

*L. 157: Write the whole name of the ratio and add (NSI) to introduce this term somewhere.*

We have followed this suggestion (line 182).

The reference of the datum has been added (line 183).

*L. 182: I would delete or reword "Similarly, a ratio between Reticulofenestra species and Florisphaera has been used to monitor changes in the nutricline and thermocline during the Pleistocene (Flores et al., 2000)." This ratio is a bit different from what was previously mentioned…*

Good point. These two ratios were not used to infer the same conditions. We have now removed this analogy.

*L. 189: The authors should acknowledge the existing differences among different proxies to reconstruct the same environmental parameter (.e.g., SSTs) and reference it. Part of the last sentence (L. 200) should be moved up in this subsection.*

This is a very valid point. The three different paleotemperature proxies have been previously used to infer paleotemperatures from various depths of the water column, from a variety of different environments and with various seasonal and geographical biases. As mentioned in our response to the "general comments", additional information has now been added in the methodology section 2.4, to explain the differences among TEX86, U37k' and Mg/Ca reconstructions and the potential biases that these introduce, as well as the reasons why we proceeded with calculating gradients between them (lines 211-241).

*This needs to be discussed further in the (sub-)section 3.4.*

The result section has been restructured according to the information that are now provided in (sub)-section 2.4. This (sub)-section (new 3.3) now mainly focuses on the results from the shelf-to-offshore paleotemperature gradients calculation, whereas the reasoning behind the different paleotemperature gradient labelling ($\Delta T$ versus $\Delta SST$) has been moved to (sub)-section 2.4 (lines 345-367).

*3. Results:*

*L. 205: How did the authors assessed the preservation? Expand or add a table in the supplementary material summarizing the ranking used. If it applies, reference it.*

Preservation in samples from U1463 and U1464 was assessed in the initial reports of Expedition 356 and followed a 1-5 scoring system (1 = poor [P], 2 = moderate, 3 = good [G], 4 = very good), following the definitions from Gallagher et al., (2017a, 2017b; Expedition 356 site summaries). In the initial reports, nannofossil preservation was assessed to score 2 and 3, demonstrating moderate and in cases good preservation. In this study, through visual inspection, we observed that good preservation was actually rare and therefore labelled the preservation for the studied

interval as moderate (score 2). We have now expanded on the matter in L.205 and have referenced the score system used (lines 245-247).

*L. 228-229: Unclear to the reader. Rephrase, please.*

We have now rephrased this sentence (lines 270-272).

*L. 234-235: "NAR of Sphenolithus spp. bounced back to higher values, especially at IODP Site U1463" This is impossible to see in Figure A2. I would recommend the authors to space out the different taxa data in Figures A1 (especially in b and d) and A2 (all of them) using different Y-axes.*

We have followed this suggestion and we now provide individual plots for each species' relative abundance and fluxes in the Appendix (Figures A1-A8), in order for the changes to be more visible.

*L. 241-242: Delete "As is the case for the nannofossil assemblage compositions", and change to: "Changes in NSI covary throughout the studied interval and correlate well between the two sites (Figure 4)."*

We have followed this suggestion (line 286).

*L. 245: change demonstrates for shows?*

We have substituted this word (line 293).

*L. 249. I suggest merging Figure 4 and 6.*

We followed this suggestion and merged Figure 4 and 6 (new Figure 4). To better accommodate this change, sub-sections 2.2 and 2.3 were also merged under the title: 3.2 *Nannofossil multivariate analysis*.

*L. 249-250: (light blue squares)… (orange triangles)*

This information has been added.

*L. 255-257. I am not fully sure I understand this sentence. Are not assemblage and species composition the same? This sounds like circular thinking, but it is probably just a matter of rewording.*

We have rephrased this sentence to read "*By color-coding the samples based on their age across the main time intervals of interest (5.4-5.2 Ma and after 4.6–4.4 Ma; Figure 5), we test if any of the observed changes in the abundance of dominant species were accompanied by shifts in the relationship between relative abundances of species in multivariate space.*"

*L. 278: Please use different symbols for the different sites.*

This figure has been replotted and merged with (old) Figure 4 (currently Figure 4b. The same symbols (but open) as in Figure 4a have been used for the different sites for consistency.

*L. 280: What does PP account for? I think it has not been introduced in the manuscript.*

PP stands for paleoproductivity. We will introduce this term in the manuscript before using the abbreviation (line 270).

*L. 285: "…Karas et al., 2011, upper…" (delete parenthesis)*

The parenthesis has been deleted.

*4. Discussion:*

*L. 326-329: Is it possible for the authors to elaborate more in these higher and lower phases?*

We have described these phases in detail in section 3.2: *Nannofossil stratification index (NSI)*

*"After ~4.2 Ma, this surface water stratification proxy shows higher variation with short intervals of higher values (4.2-4 Ma and 3.8-3.54 Ma), that correspond to peaks in relative abundance of Sphenolithus spp. Between 4 and 3.8 Ma, NSI demonstrates low values…"*

This sentence will be written more concisely to read: ""*After ~4.2 Ma, NSI shows higher variation with short intervals of higher values (4.2-4 Ma and 3.8-3.54 Ma). These intervals correspond to peaks in relative abundance of Sphenolithus spp.*"

Additionally, in the discussion, we now elaborate more on the possible reasons that might have led to higher variability between higher and lower phases of NSI during this interval. Although the mechanism is unclear, it is interesting that higher amplitude NSI after 4.2 Ma, correlate well with higher amplitude changes observed in the temperature gradients after ~4.3 Ma. This would suggest a similar controlling mechanism to the one we are proposing in this manuscript, but in this case on shorter timescales (lines 391-395; lines 447-451).

*L. 345: "shelf area due to…" (I would mention it in the first sentence or merge L. 344-347).*

We have followed this suggestion and merged L. 344-347 (lines 410-412).

*L. 360: What about a combination of two or three of the 3 proposed mechanisms? Would that be an option? If so, perhaps add a sentence.*

We have added a sentence mentioning that any possible combination of the proposed mechanisms could have had similar results (lines 426-428; 475-476).

*L. 371-372: "Shift" is mentioned twice in a sentence. Find a synonym.*

The first "shift" has been replaced with "change" (line 439).

*L. 420-470. The reference Stuut et al. (2019) (https://doi.org/10.1029/2019GL083035) could be useful in the section 4.3 from the discussion because that study (in the continental margin of NW Australia) covers the last 5.3 Ma.*

This reference has been added in the section 4.2, as part of the overall paleoclimatic regime in the continental margin of NW Australia during the early Pliocene before we focus on paleoproductivity and paleoecology matters (line 444).

*L. 423: Make sure PP is introduced before (not just in the figure captions), I guess the first time is in L. 67.*

This abbreviation has been introduced in the text (line 270).

*L. 478: Trilobatus sacculifer*

We have written the full name of the genus across the manuscript.

*L. 534-547: Figures A1 and A2. I already mentioned it, but I think it is difficult to see the data in some of the plots from these figure (especially if you print it in black and white). I would suggest using different Y axis.*

As mentioned earlier, we have replotted the Appendix figures with one species in each panel so that the data are more visible to the reader.

*When referring to a specific taxa, please add the symbol in the caption on top of the colour, as it was done for example in Figure C1. E.g., (<5μm Reticulofenestra and <3μm Gephyrocapsa; green circles).That will help colour-blind readers.*

We have followed this suggestion.

*Also, I would probably include them as main figures (not as appendix), but this is up to the authors.*

We considered this suggestion, although we think that the figures contain a lot of information (as you probably already noticed), which are not all presented in detail in the main manuscript and would be tangential to the main focus of this manuscript. We would like to keep a "grouping-oriented" strategy by focusing mainly on dominant species and ratios between them. Yet, we want to show all (raw) data that is included in the CA and Shannon-index calculations for transparency and for those seeking more details, in the supplement.

*L. 537: Rather than error bars, I would use "shaded areas"*

Good point. We now use the term "shaded areas" in the captions.

*L. 545: 15%; Bordiga ... (delete parenthesis).*

This parenthesis has been deleted.

*In Figs. B1 and C1 the symbols of the legend do not match with the ones in the plot.*

Good observation, thanks. We have corrected the legend symbols in Figures 9, 10, so that they match the plots, as well as replaced the symbols with the ones used in the main figures of the manuscript.

*References:*

*There are several typos in the reference list. I spotted few, but the authors need to carefully check all the references.*

*L. 589: 2.45 Ma (space missing)*

*L. 594: (80-. )., (revise, something is missing here)*

*L. 595: K.-H.*

*L.598: $CO_2$ (subscript)*

*L. 606: 7(May) Double check*

*L. 614: Holloway*

*L. 626: 925, and… (delete comma?)*

*L.631: (August) Double check*

*L. 633: (80-. )., (revise, something is missing here)*

*L. 653: ,, (delete one comma)  (February) Double check*

*L. 657: Species names' in italics*

*L. 665-666: Add doi*

*L. 672: PAST*

*L. 675: CO$_2$ (subscript)*

*L. 684-686: Add doi*

*L. 693-694: Add doi*

*L. 695: Nye, H. ..: (delete one dot)*

*L. 707-714: B-Th, B. –. T. or B. T.? The name of the main author is written in 3 different ways.*

*L. 756: Page numbers or number of pages?*

*L. 769: (80-. )., (revise, something is missing here)*

We thank the reviewer for taking the time to identify these typos. The references were revised accordingly.

References

Gallagher, S.J., et al., 2017. Site U1463. In Gallagher, S.J., Fulthorpe, C.S., Bogus, K., and the Expedition 356 Scientists, Indonesian Throughflow. Proceedings of the International Ocean Discovery Program, 356: College Station, TX (International Ocean Discovery Program).

Gallagher, S.J., et al., 2017. Site U1464. In Gallagher, S.J., Fulthorpe, C.S., Bogus, K., and the Expedition 356 Scientists, Indonesian Throughflow. Proceedings of the International Ocean Discovery Program, 356: College Station, TX (International Ocean Discovery Program).

**Author's response (R2)**

Karatsolis et al. – cp-2022-60

We found this review constructive and helpful, and we therefore want to thank Anonymous Referee #2 for taking the time to read our work and suggest changes that will make the analysis more robust and the discussion clearer.

*Comment on cp-2022-60*

*Anonymous Referee #2*

*Referee comment on "Late Neogene nannofossil assemblages as tracers of ocean circulation and paleoproductivity over the NW Australian shelf" by Boris Theofanis*

*Karatsolis and Jorijntje Henderiks, Clim. Past Discuss., https://doi.org/10.5194/cp-2022-60-RC2, 2022*

*The Karastolis and Hendericks study is an interesting contribution in a region and period that need to improve knowledge, particularly.*

*The use of CN offers a unique opportunity to characterize surface water masses and to monitor their evolution during the Mio-Pliocene transition interval.*

*The state of the art is well stablished and objectives of interest.*

We thank the reviewer for the positive comments.

*The chosen technique is appropriate, based on previous initiatives. The use of sphenoliths as main taxonomic group linked to stratification is fine, although should be more correct to include too discoasters (although its proportion is low). Others species clod be also considered in the interpretation. In fact there are some that are present in the plots but not mentioned/considered. If they are not taken into account, there is no point in including them.*

Discoasters have been used to infer similar (more oligotrophic and stratified) conditions to those we are reconstructing in this study using *Sphenolithus* spp. (e.g., Imai et al., 2015). However, in this case, the relative abundances of *Discoaster* spp. are too low to be included in our analysis, which focuses only on common and dominant species. As an indication, Discoasters never exceed 1% of the assemblage in samples from Site U1463 and only do so in 2/105 investigated samples from Site U1464, in which case they still remain under 2%. This indicates, most importantly, that including Discoasters in the analysis would not alter the results and the interpretations we make. The relative abundances and fluxes of the rest of the common species (*Calcidiscus* spp., *Umbilicosphaera* spp., *Reticulofenestra* spp. >5μm) do help us understand better the extent of the changes in the nannofossil assemblage across the two intervals of interest (5.4-5.2 Ma; 4.6-4.4 Ma), as well as to put them in broader-scale context. See discussion in section 4.3: *Broader-scale changes in paleoproductivity and paleoecology* (L.514-519).

*One aspect repeatedly used is seasonality. In the text it is discussed that with this resolution a direct correspondence with present day conditions it is not feasible.*

*In this sense, should be more correct talking about persistent conditions (e.g. stratification), rather than enhance seasonality.*

*In fact, the signal that is manifested is the dominant one in a sufficiently broad period to refer its dynamics to seasonality (although it is obvious that these processes may be the triggers), but there are not enough arguments to focus these changes on seasonal variation only.*

*I suggest a modification in this sense, avoiding such a direct reference to seasonality.*

This is an interesting and valid point. Indeed, the timescales that we study make it impossible to directly infer and resolve seasonal variations, as we state in the manuscript. However, sustained changes in seasonal intensity could have had a long-term impact in regional ocean circulation and paleoclimate that, in turn, can be recorded in the sedimentary archive. For example, although we cannot resolve the intensity of the Leeuwin current (LC) during an austral winter thousands of years ago, an overall increase of sea surface temperatures (SST) offshore west Australia (LC pathway) can be used as an indirect way to infer its presence. In turn, since this current flows seasonally, we could propose that this SST change could be linked to a stronger winter season that could have intensified the flow of the LC. That being said, we understand the concern of building our argument primarily around seasonality and making it the main focus point of the proposed mechanism. To resolve this, we have now toned down our claims regarding a direct link to changes in seasonality on several occasions, and we mainly focus more on the long-term changes in stratification and water column mixing, that can be substantially supported by the provided data (e.g., lines 19, 25, 457, 470, 495, 601).

*Other aspect that must be considered in Discussion and Conclusions, is provide more clear information about the ITF and LC, marked as objective, trying to ling the signal observed with characteristics of LC (for example). In fact this is considered, but plots should be explained in this sense, in order to follow better the arguments. Discussion refers the features in a general way without focusing on the evolution of those water masses. In this sense, I cannot see clearly the utility of the H Index (not linked to these features) and the CA (also superfluous. With the available data (T) and stratification index, should be possible to identify/define the involved water-masses, and consequently its evolution along the considered period.*

We have rewritten parts of the Discussion to better explain the suggested changes in LC activity, as well as the possible link of their features to the temperature gradients and NSI signals we identified (e.g., lines 455-461, 465, 478-479, 470-473). Details regarding the shift from Regime 1 to Regime 2 have also been added in the caption of Figure 7, in order to better link this figure with the text (lines 568-570).

As for the H-index and the correspondence analysis, indeed, they are not directly used to infer changes in stratification. However, they are useful in providing additional information regarding the broader-scale (and persistent) changes in paleoproductivity and paleoecology and therefore help us understand the global-scale processes recorded in the nannofossil assemblage. (see section 4.3 *Broader-scale changes in paleoproductivity and paleoecology*; L.504-506, 514-516). For this reason, we believe that they should be presented as part of the results and included in the discussion.

*Concerning the potential mechanisms related to paleoenvironmental aspects (3 options), the authors should consider the most reasonable possibility, taking into account the available data.*

It is not entirely clear what the reviewer is suggesting in this sentence. The three possible mechanisms/ocean circulation scenarios are presented in section 4.1: *Water column mixing and nutrient availability on the NW Australian shelf*, based initially on the changes in the NSI. Later, after also discussing the observed changes in the temperature gradients in the first part of section 4.2: *Paleotemperature and inferred ocean circulation patterns*, we present the most reasonable (to our understanding) scenario/combination of features, taking into consideration all the available information (L.469-483):

The three mechanisms that could have led to sustained changes in NSI are summarized here to facilitate the revision:

1. If the LC activity had intensified, it could have led to increased eddy formation that promoted enhanced productivity across the western Australian continental shelf.

2. An overall increase in convective mixing could have occurred across the continental shelf area due to more cooling in the upper water column and/or intensified storm activity during the winter period.

3. If the oligotrophic influence of the LC (which is understood as: warm, oligotrophic LC waters can inhibit upwelling activity and therefore lead to overall decreased primary productivity across the continental shelf) had weakened, productivity over the shelf may have had increased.

A sentence that links our hypothesis to these three mechanisms, and then explains which one we see as the most reasonable scenario according to our analysis, has been added. The possibility that a combination of mechanisms was present (also suggested by Reviewer 1) is now discussed (lines 426-428, 475-476).

*The comments referred in line 395 is too speculative: should provide extra information to propose this mechanism, or afford the explanation in a more general way (mixing!).*

We have rewritten this part of the discussion to better explain our arguments and link the combination of observations in NSI and temperature gradients to the mechanisms we described in section 4.1 (lines 469-479).

*Section 4.3 refer an interesting global feature. Here is considered lightly, being necessary a better explanation of the processes and records. The link with the rest of the text is not clear, need better justification.*

We have added additional information in this section to further explain the processes behind broader-scale changes in calcareous nannofossil paleoproductivity and paleoecology (lines 519-526). In the previous sections, we focused on regional paleoclimatic and ocean circulation changes and how these are recorded in the nannofossil assemblage. In this section, we aim to investigate how broader-scale changes in paleoproductivity are reflected in the assemblage and therefore decouple them from the more regional ones. A sentence linking this section to the rest of the discussion and pinpointing the importance of looking at the bigger picture has been added in the beginning (lines 498-500).

References

Imai, R., et al., 2015. Evidence for eutrophication in the northwestern Pacific and eastern Indian oceans during the Miocene to Pleistocene based on the nannofossil accumulation rate, *Discoaster* abundance, and coccolith size distribution of *Reticulofenestra*, Mar. Micropaleontol. 116, 15–27.